# CRSA: A Chinese Single-Domain Task-Oriented Dialogue Dataset with Contextual Rich Semantic Annotations

## ABSTRACT

Task-oriented dialogue (TOD) systems support users in achieving domain-specific goals via natural language interactions and critically depend on high-quality datasets. However, existing datasets often lack authenticity, fine-grained semantic annotations, and explicit process control, limiting effectiveness in complex business scenarios. To address these, we introduce CRSA, a Chinese TOD dataset that integrates diverse sources to construct semantically rich, structurally realistic dialogues, and adopts a multi-level annotation framework to model dialogue acts, user intents, and task flows more effectively. To evaluate the quality and application potential of CRSA, we conduct three sets of experiments spanning data quality, system training effectiveness, and task adaptability. Results demonstrate that CRSA provides strong support for process modeling, strategy learning, and response generation, establishing it as a robust and versatile resource for TOD research. The dataset is publicly available at `https://anonymous.4open.science/r/CRSA-CBBB`.

## 1 INTRODUCTION

Research on building TOD systems has become a key focus in natural language processing (Zhang et al., 2020), driven by their practical value in real-world applications (Ling et al., 2023). Despite progress has been made, most existing systems often prioritize broad multi-domain coverage (Su et al., 2021) and information retrieval (Valizadeh & Parde, 2022), yet lack effective business modeling and behavior control, limiting their ability to maintain awareness dialogue flow awareness and support structured interactions in real scenarios (Prajapat & Toshniwal, 2024).

Current TOD research emphasizes subtask performance(Zhu et al., 2024), yet struggles with vague expressions, semantic ambiguities, and atypical user behaviors—particularly in context-dependent, domains(Zhu & Xu, 2025). Existing TOD datasets further limit development due to simplistic structures, coarse-grained annotations, and insufficient coverage of dialogue behaviors(Valizadeh & Parde, 2022), hindering fine-grained modeling and semantic understanding(Feng et al., 2023).

To address the aforementioned challenges, we construct a new multi-turn TOD dataset targeting real-world complex business scenarios—a class of tasks characterized by tightly coupled constraints, interdependent decision factors, and dynamic user requirements (formal definition in Section 3.1). Unlike datasets that emphasize domain or language expansion (Zhao et al., 2024; Algheairy & Ahmed, 2024), we focus on the airline booking scenario, a representative instance involving multi-constraint reasoning, incrementally disclosed user needs, evolving preference patterns, and diverse interaction contingencies. These properties reflect the operational complexity of real-world service dialogues and highlight the need for more structured, process-aware TOD modeling.

This paper introduces an optimized pipeline for constructing TOD datasets, encompassing data collection, processing, and annotation. We propose a construction strategy that emphasizes structural integrity, fine-grained semantic supervision, and process controllability, culminating in the creation of high-quality CRSA dataset. Comprehensive experiments are conducted on dataset quality evaluation, analysis of its effectiveness in supporting model training and multi-task adaptability, demonstrating the substantial contribution of CRSA to the development of TOD system. The main contributions of this work are summarized as follows:

- A dialogue corpus construction methodology tailored for complex business scenarios is proposed, encompassing multisource data integration, interaction flow design, and structural standardization strategies to improve semantic complexity and pragmatic coverage.

- We design a fine-grained hierarchical annotation framework covering key dimensions. Built upon this framework, we construct CRSA, the first Chinese TOD dataset explicitly designed for process modeling and controllable response generation, providing high-quality support for dialogue system studies and real-world applications.

- We conduct comprehensive evaluations from multiple perspectives. Results consistently demonstrate CRSA's effectiveness in supporting robust TOD system development and its potential as a challenging benchmark for future multi-task dialogue research.

## 2 RELATED WORK

TOD systems are mainly developed using two approaches: modular pipeline and end-to-end generation. Pipeline methods (Ohashi & Higashinaka, 2023; Zhu et al., 2019) model natural language understanding (NLU), dialogue state tracking (DST), dialogue policy(DP), and natural language generation (NLG) separately, offering flexibility but facing error propagation (Qin et al., 2023) and joint optimization challenges (Liu & Lane, 2018). Recent studies explore mitigating these issues via unified or generative modeling (Tseng et al., 2020), yet many real-world dialogue situations remain insufficiently represented and lack well-defined handling strategies. End-to-end methods (Li et al., 2024) integrate all components within a single framework, simplifying development.

High-quality data resources is essential for advancing TOD system performance. Multi-WOZ (Budzianowski et al., 2018) is a widely used TOD datasets, covering various scenarios and supporting research in subtasks. However, it suffers from annotation inconsistencies and limited coherence (Kulkarni et al., 2024). SGD (Gower et al., 2019) introduces structured schemas to drive dialogues and emphasizes domain expansion and zero-shot generalization. While enabling generalization, its loose structure and weak context limit dialogue flow control.

In TOD research, several representative datasets and resources have laid the foundation for scalable interactive modeling, including CrossWOZ (Zhu et al., 2020), RiSAWOZ (Quan et al., 2020), and CGoDial (Dai et al., 2022). AirDialogue (Wei et al., 2018) provides a large goal-oriented benchmark for travel planning, while ConvLab-3 (Zhu et al., 2023) offers a unified framework that integrates heterogeneous datasets and supports modular system development.

However, existing datasets and frameworks still fall short in representing dynamic user behaviors, atypical expressions, and system-driven flow control required by high-complexity real-world scenarios. Most rely on constrained response patterns or limited annotation granularity, restricting their scalability in process modeling, exception handling, and proactive strategy generation. In contrast, the contribution of CRSA lies in proposing a transferable deep semantic annotation methodology that can generalize to arbitrary complex business workflows.

## 3 DATASET

### 3.1 DATA COLLECTION

Complex business scenarios, as considered in this work, refer to task settings characterized by: (i) multiple interdependent operational constraints; (ii) user requirements that are disclosed incrementally across turns; (iii) context-dependent adjustments of feasible options; (iv) interaction patterns that introduce task deviations, revisions, or embedded subtasks; and (v) explicit business procedures that restrict allowable system actions. These properties collectively impose higher demands on flow modeling and decision consistency in TOD systems.

Airline booking serves as a representative instance due to its multi-constraint decision structure and multi-turn requirement elicitation. Grounded in this setting, we construct the CRSA dataset to address limitations of existing TOD resources in semantic diversity and process-level supervision. CRSA integrates three complementary sources—real business dialogues, crowd-sourced simulations, and LLM-assisted generation—to ensure authenticity, coverage, and diversity.

In the initial phase, real-world dialogues were collected, covering the full process from user inquiry to business completion. After transcription, anonymization, and semantic cleaning, high-quality samples were retained as semantic foundation of the dataset.

To expand data volume and semantic coverage, we conducted crowdsourced paired simulations guided by two protocols— **System-side Dialogue Behavior and Response Strategy Specification** (Appendix B.1) and **User-side Requirement Expression and Interaction Process Protocol** (Appendix B.2). These protocols ensure behavioral consistency and structural complexity. System-side workers controlled dialogue flow and handled exceptions, while user-side participants produced diverse, multi-path requests to increase semantic and behavioral variability.

For LLM-generated dialogues, we adopt a GPT-4o (Hurst et al., 2024) few-shot framework that composes task descriptions, exemplars, and behavioral constraints into a three-layer prompt, augmented with domain knowledge to improve coherence and compliance. Quality is ensured via semi-automatic filtering: system checks for structural integrity and slot consistency, followed by expert review of semantic coherence and style.

CRSA integrates data from the three aforementioned sources, ensuring semantic depth and task coverage. All data are stored in a multi-turn dialogue format to facilitate downstream processing.

## 3.2 DATA PROCESSING

Following multi-source data collection, we conducted systematic cleaning and structural normalization to ensure consistent quality and modeling value. The processing pipeline includes three main stages: data cleaning, system-led structural standardization, and dialogue history modeling.

During cleaning, we removed dialogue-level samples that were unusable for modeling, including cases with severe transcription errors that rendered the conversation incomprehensible, and instances where missing critical turns prevented reconstruction of the underlying business flow.

Each data modeling unit consists of the full dialogue history preceding a system response, structured into three stages: basic information collection, candidate selection, and task finalization. This structure captures task progression and user intent evolution, while forming clear semantic segments to support multi-turn context modeling, state tracking, and strategy learning.

To enhance the representation and modeling of dialogue flow control, we refine semantics and embed control strategies during data processing, reinforcing the system's leading role. We establish a standardized, stage-wise progression mechanism grounded in business logic, specifying prompt order and guidance behaviors at each stage. System utterances lacking pragmatic clarity or sufficient guidance are supplemented or rewritten. We also integrate response strategies for atypical interactions, including redirection for off-topic inputs and clarification for vague requests. All system responses are reviewed and revised to ensure contextual coherence and effective flow control. These revisions increase semantic granularity and diversity across interaction patterns and task stages.

As a result of the above processing, we construct a TOD corpus with structured logic, coherent semantics, and explicitly defined system behaviors.

## 3.3 DATA ANNOTATION

Building on standardized data quality and dialogue flow, we construct a multi-level semantic annotation framework to support system modeling and user understanding in complex task dialogues.

The annotation schema comprises three tiers—**Context**, **Dialogue**, and **Slots**. **Context** captures dialogue history, system control actions, and user goals to reflect state progression. **Dialogue** labels current-turn system intent and user response and explicitly marks anomalous user behaviors. **Slots** aggregates global slot values and tracks state updates across turns. These layers jointly build an integrated representation of semantics, behavior and state.

To enhance robustness to non-canonical and unstable user behaviors, CRSA introduces a systematic anomaly modeling mechanism. Six anomaly types—covering goal shifts, off-topic turns, and conflicting or incomplete constraints—are explicitly annotated (Table 1). Coupling these labels with stage logic and system actions provides supervision for recovery strategies, enabling models to main-

Table 1: User anomaly taxonomy and descriptions

| Anomaly type | Description |
| --- | --- |
| **Unclear** | Ambiguous or undefined slot reference |
| **Default** | Request for system recommendation |
| **Vague** | Imprecise slot value affecting state tracking |
| **Alter** | Modification of a previously filled slot |
| **Irrelevant** | Utterance unrelated to the current task |
| **Error** | Input with logical inconsistency or factual error |

tain process correctness under noisy or drifting inputs. System responses are further decomposed into **descriptive feedback** and **progressive inquiry** for aligned exception handling.

System behavior is annotated using a triplet structure: **dialogue act + query operation/slot + associated keys**, combining execution logic and semantic intent. The set of behavior includes 63 types that cover task guidance, recommendations and recovery strategies corresponding to deviations.

We further introduce controllable behavioral labels that encode system style, pacing, subtask handling strategies, and responses to non-task queries, enabling consistent and user-tailored generation.

For slot annotation, CRSA introduces two mechanisms to handle fuzzy and subjective expressions. **Slot value normalization** maps vague inputs to standard ranges or categorical labels, allowing the model to interpret open-ended preference expressions while appropriately grounding them in the structured constraints required for downstream decision making. **Subjective slot mapping strategy** assigns intermediate semantic tags to user preferences, leveraging context to constrain candidate values and guide personalized recommendations.

To enable large-scale annotation, we train an automatic annotation model based on Baichuan2-7B (Yang et al., 2023), optimized under a multi-task learning framework. The primary task is generating the structured annotation, supported by auxiliary tasks including anomaly detection, behavior classification and state tracking. Each task is handled via an independent prediction head, and task-specific representations are extracted through a lightweight adaptation module defined as:

$$h_{\text{task}} = h_{\text{shared}} + W_{\text{up}} \cdot \text{ReLU}(W_{\text{down}} \cdot h_{\text{shared}})$$

where $W_{\text{down}}$ and $W_{\text{up}}$ are projection matrices that reduce and recover feature dimensionality, enabling parameter efficiency and residual learning.

To improve the model's capability for minority classes and complex distributions, a joint loss function integrating multiple loss terms with dynamic weighting is proposed:

$$\mathcal{L}_{\text{total}} = \alpha \cdot \mathcal{L}_{\text{CE}} + \beta \cdot (\mathcal{L}_{\text{hinge}} + \lambda \cdot \mathcal{L}_{\text{focal}}) + \gamma \cdot \mathcal{L}_{\text{KL-div}}$$

The weights $\alpha, \beta, \lambda, \gamma$ are tuned via development set, enabling the model to dynamically balance tasks of varying granularity and difficulty.

Subsequent experiments (Section 4.1.3) validate the adaptability of this model. This design decomposes stage-level semantic reasoning, behavioral linkages, and flow–decision dynamics in business dialogues into a learnable annotation framework, forming a deep semantic labeling methodology that can be transferred to arbitrary complex business domains.

### 3.4 STATISTICS

All high-level indicators and comparative advantages of CRSA are summarized in Table 2. The dataset comprises three complementary sources—25.1% real interactions, 54.6% human role-playing dialogues, and 20.3% GPT-4o–augmented conversations—which collectively introduce substantial behavioral heterogeneity. The corpus exhibits a high anomaly rate of up to 39.2%, with frequent stage regressions (27.9%) and subtask insertions (33.3%), reflecting the ambiguity, discontinuity, and goal-shift phenomena characteristic of real task-oriented interactions. These properties result in a broad-coverage and controllably heterogeneous dataset that better captures the irregularities encountered in real-world dialogue systems.

Table 2: Statistical comparison of crsa with other datasets in tod systems

| Dataset | MultiWOZ | RiSAWOZ | CrossWOZ | SGD | CRSA (ours) |
|---|---|---|---|---|---|
| Language | en | zh | zh | en | **zh** |
| Dialogs per domain | 1205 | 934 | 1002 | 1008 | **1480** |
| Turns | 16222 | 11215 | 16938 | 20622 | **26048** |
| Avg. Turns | 13.5 | 12.0 | 16.9 | 20.4 | **17.6** |
| Avg. Slots | 9.4 | 13.25 | 14.4 | 13.65 | **15** |
| Avg. Values | 956.6 | 861.4 | 1574.2 | 883.7 | **1713** |

Beyond user behavior, CRSA also provides rich supervision for modeling system actions and dialogue controllability. The system employs 39 distinct questioning strategies, 16 types of query responses, and 7 exception-handling strategies, offering fine-grained pragmatic signals for policy learning. User utterances display natural variability, with 31.4% involving indirect expressions and 26.9% containing out-of-domain or anomalous content. The dialogue flow is predominantly system-led: 83.7% of user deviations are actively redirected, 62.5% of key slots are completed through system-initiated prompts, and over half of stage transitions are proactively triggered by the system.

## 4 EXPERIMENT

### 4.1 EVALUATION OF DATA AND ANNOTATION

#### 4.1.1 DIALOGUE CORPUS EVALUATION

This experiment assesses the corpus quality of CRSA—focusing on semantic complexity, contextual modeling difficulty, and system learning performance. For comparability, CRSA is converted to the RiSAWOZ format and annotation schema and trained with mBART (Chipman et al., 2022); several mainstream TOD datasets are evaluated under the same configuration. We report DST Accuracy, DA Accuracy, BLEU, and ROUGE-L (definitions in Appendix C).

Table 3: Performance comparison across datasets. [†]: significant difference ($p < 0.05$).

| Dataset | DST Acc. | DA Acc. | BLEU | ROUGE-L |
|---|---|---|---|---|
| MultiWOZ | 84.7 | 84.2 | 20.7 | 40.2 |
| SGD | 87.9 | 82.3 | 24.6 | 43.8 |
| RiSAWOZ | 82.5 | 78.3 | 23.2 | 32.7 |
| CrossWOZ | 84.1 | 85.4 | 21.5 | 34.5 |
| TransferTOD | 79.4 | 86.7 | 28.5 | 31.9 |
| **CRSA (ours)** | **72.9**[†] | **76.1**[†] | **14.6** | **20.9** |

As shown in Table 3, CRSA scores lower across all metrics. Lower DST accuracy indicates more variable slot mentions and less repetitive slot filling; lower DA accuracy reflects more diverse system behaviors, making act prediction harder; and lower BLEU or ROUGE-L suggest fewer templated responses and greater linguistic variability.

These results demonstrate that conventional shallow annotation schemes are insufficient to model the semantic phenomena and structural variability in CRSA, underscoring the need for a more expressive, process-aware framework. They further show that the real-world dialogue phenomena challenge existing annotation paradigms, particularly in handling fine-grained interaction dynamics.

#### 4.1.2 EFFECTIVENESS OF THE ANNOTATION SCHEME

This experiment evaluates the impact of the CRSA annotation framework on corpus utility and model performance. To isolate the effect of annotation, multiple Chinese TOD datasets are evaluated under two conditions: (i) their original annotations and (ii) re-annotation aligned to the CRSA scheme. For both settings, we train the same mBART model with identical hyperparameters.

Table 4: Performance gains with CRSA annotations. $^\dagger$: significant at $p < 0.05$.

| Dataset | $\Delta$ DST Acc.$^\dagger$ | $\Delta$ DA Acc.$^\dagger$ | $\Delta$ Intent Acc.$^\dagger$ |
|---|---|---|---|
| CrossWOZ | +3.9 | +1.7 | +2.5 |
| RiSAWOZ | +4.8 | +3.3 | +3.8 |
| TransferTOD | +2.7 | +2.4 | +1.9 |

As shown in Table 4, CRSA-style re-annotation yields consistent improvements across datasets. We attribute these gains to the additional semantic signals introduced by the new annotation scheme, which make latent contextual and procedural information explicitly accessible to the model.These findings validate the effectiveness of the proposed scheme in enhancing corpus annotation quality and supporting downstream model training.

### 4.1.3 ANNOTATION QUALITY AND EFFICIENCY

To evaluate the annotation performance and efficiency of the human-machine collaborative annotation framework adopted in CRSA, we compare manual and model-generated annotations. Using the model from Section 3.3, we sample 300 unseen, unannotated dialogues from each data source and annotate the same samples with both methods. Four fields—`current_step`, `anomaly_analysis`, `agenda`, and `Slots`—are assessed using exact-match accuracy.

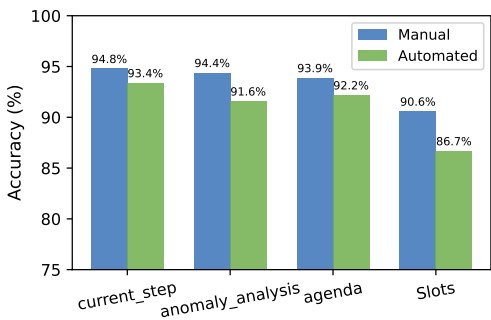

Figure 1: Accuracy comparison of two annotation paths

Table 5: Comparison of manual and automated annotation efficiency and quality

| Metric | Definition | Manual | Automated |
|---|---|---|---|
| AAT (sec/case) | Avg. time per annotated dialogue | 57.6 | **1.72** |
| Revision (%) | Proportion of cases requiring edits | 6.2 | 11.3 |
| FMI (fields/case) | Avg. fields revised per dialogue | 0.17 | 0.68 |

Figure 1 shows that both methods exceed 86% accuracy across all fields. Table 5 indicates that automatic annotation markedly reduces processing time, with a moderate increase in revision rate and fields modified. These results demonstrate that the CRSA annotation framework ensures high structural and semantic consistency while offering strong learnability and efficient scalability. The automatic model achieves accuracy comparable to manual annotation, validating the feasibility and practicality of the proposed CRSA pipeline for large-scale TOD datasets construction.

### 4.1.4 ABLATION STUDY

We conduct ablation experiments following standard TOD evaluation practice, comparing both conventional metrics (DST/DA/API, BLEU) and process-level indicators reflecting dialogue flow and task completion (TCR, SAR, EDR). Details are provided in Appendix C.2.

Table 6: Performance comparison of ablation experiments ("–" indicates the ablated annotations)

| Ablation Configuration | DST | DA | API | TCR | BLEU | STA | EDR |
|---|---|---|---|---|---|---|---|
| Original Annotation | **80.3** | **83.6** | **89.2** | **85.1** | **26.9** | **89.4** | **91.3** |
| – Extended Dialogue Act | 78.6 | 70.1 | 82.5 | 75.4 | 17.2 | 76.3 | 86.4 |
| – User Response Anomaly | 79.2 | 75.2 | 84.8 | 68.7 | 26.2 | 72.4 | 90.4 |
| – Alternative Options | 79.8 | 81.4 | 88.2 | 61.3 | 15.3 | 84.1 | 89.7 |
| – Stage-Based Annotation | 69.3 | 74.3 | 86.3 | 83.2 | 26.7 | 74.9 | 86.5 |
| – Query Operation Tags | 76.6 | 78.7 | 73.5 | 76.2 | 33.5 | 86.3 | 90.4 |

As shown in Table 6, ablating any component degrades performance, demonstrating the distinct contribution of each. Removing extended dialogue acts causes the largest drops in DA Acc. and BLEU, reflecting their role in response generation and act prediction. Omitting user anomaly analysis lowers TCR and STA, reducing robustness to non-canonical inputs. Excluding alternative-option records markedly reduces TCR and BLEU, impairing multi-turn decision modeling. Dropping stage-based labels harms DST Acc. and STA, weakening process/state grounding. Eliminating query-operation tags most severely affects API Acc., confirming their necessity for execution accuracy. These findings confirm that CRSA's layered, task-aware annotations are essential for flow control, exception handling, and intent alignment, validating its fine-grained and task-aware annotation strategy.

## 4.2 Evaluation of TOD System Training

### 4.2.1 Comparative Analysis of Dataset Training Effectiveness

CRSA is developed to provide high-quality training data for TOD systems in real-world scenarios. We assess CRSA's training effectiveness by fine-tuning Baichuan2-7B separately on CRSA, Cross-WOZ, RiSAWOZ, and TransferTOD under an identical pipeline (same preprocessing, hyperparameters, and procedures). Evaluation uses a 500-sample multi-turn test set drawn from each dataset's test split. We report standard dialogue metrics plus two process/pragmatics measures—ADFC (flow and slot control) and CRAM (contextual appropriateness; definitions in Appendix C.3).

Table 7: Model performance comparison across TOD datasets. [†]: significant at $p < 0.05$.

| Metric | CRSA (Ours) | CrossWOZ | RiSAWOZ | TransferTOD |
|---|---|---|---|---|
| Intent Acc. | **92.6**[†] | 86.9 | 87.3 | 90.6 |
| Action F1 | **93.5**[†] | 90.2 | 88.2 | 90.1 |
| TCR | **92.8**[†] | 83.2 | 83.5 | 87.4 |
| BLEU | 29.3 | 37.4 | 36.1 | **39.6** |
| Distinct-2 | **39.2** | 32.6 | 31.9 | 36.1 |
| ADFC | **0.89**[†] | 0.81 | 0.75 | 0.84 |
| CRAM | **0.87**[†] | 0.74 | 0.79 | 0.81 |

As shown in Table 7, the model fine-tuned on CRSA achieves strong overall performance across multiple metrics, demonstrating its effectiveness in task flow control, dialogue strategy generation, and contextual adaptation. BLEU is slightly lower than TransferTOD, consistent with CRSA's less-templated language and higher lexical variability.

Empirical results underscore the strengths of CRSA's structured annotation and its coverage of complex interaction—including user anomalies, flexible slot filling, and system-led control mechanisms. In contrast to more rigid or template-based datasets, CRSA provides rich supervision for training dialogue models, making it a robust training resource for developing high-quality TOD systems.

### 4.2.2 CONTROLLABILITY AND FLOW AWARENESS

This section evaluates whether CRSA-trained models exhibit *Process Awareness*—inferring the current task stage and producing logically appropriate responses without explicit prompts (Wu et al., 2021)—and *Controllability*—consistent adherence to external control signals (Liang et al., 2024).

Table 8: Control dimensions, goals, and example tokens for controllable generation

| Dimension | Goal | Example Tokens |
|---|---|---|
| Response Style | Regulate tone of reply | <kind / neutral / blunt> |
| Deviation Handling | Manage off-task user input | <reject / skip / redirect> |
| Query Strategy | Control detail level | <brief / detailed / ignore> |
| Flow Strategy | Enforce slot filling order | <slots-seq-i> |

We fine-tune Baichuan2-7B-Chat using annotated dialogue history with optional control tokens. Control dimensions cover response style, deviation handling, query strategy, and flow strategy (Table 8). Responses are generated on a 200-sample multi-turn test set. Evaluation combines *process awareness* metrics—SCR (slot–inquiry alignment) and PAE (contextual suitability of advancement)—and *controllability* metrics—SCC, DHC, QRSC, and FCC (definitions in Appendix C.4).

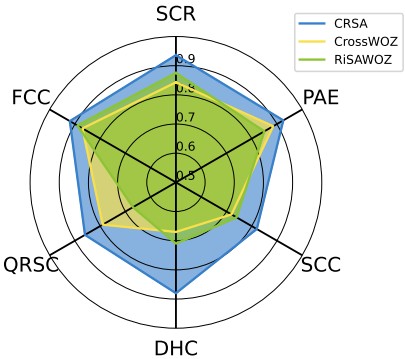

Figure 2: Flow awareness (SCR, PAE) and controllable generation (SCC, DHC, QRSC, FCC) results

As shown in Figure 2, CRSA-tuned models outperform baselines on both SCR and PAE. For controllability, it consistency exceeds 87% across all dimensions, reliably following control tokens to produce conditioned responses. Taken together, these gains indicate that CRSA's stage- and behavior-aware supervision yields stronger dialogue-flow control and stable response conditioning, substantiating its advantage for training controllable TOD systems.

## 4.3 MULTI-TASK ADAPTABILITY OF CRSA

### 4.3.1 BENCHMARKING TOD SUBTASKS

We benchmark CRSA's adaptability on four canonical TOD subtasks against mainstream TOD datasets. mBART is used for NLU/DST/DP subtasks and Baichuan2-7B-Chat for NLG. All datasets are standardized to a common format (slot–value pairs, dialogue-act labels, target texts). Training strategies and hyperparameters are held fixed. Each task uses equal-sized, independently built train splits; test sets are uniformly formatted and source-balanced to reduce dataset bias.

Figure 3 reports subtask accuracy, and Figure 4 reports NLG quality and diversity. Under the same model and training setup, CRSA exhibits lower NLU, DST, and DA performance despite sharing a unified evaluation schema with other datasets. Combined with our statistical observations on anomaly frequency, enlarged state space, linguistic variability, and process-control difficulty, these results indicate that CRSA imposes substantially higher demands on semantic parsing, expression grounding, exception handling, and flow modeling.

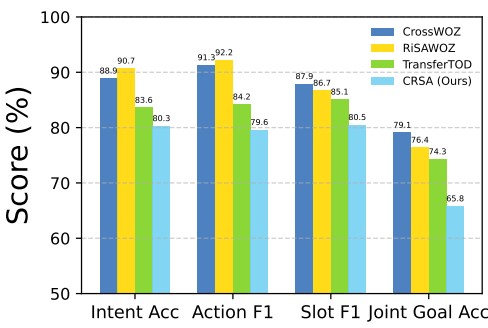

Figure 3: TOD Subtask Accuracy

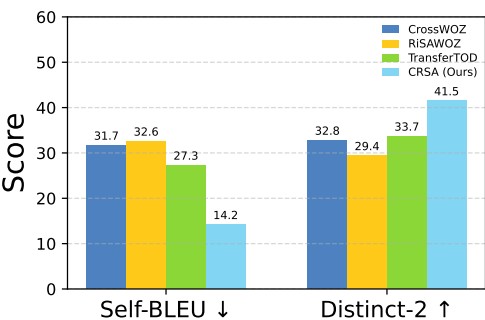

Figure 4: NLG Quality and Diversity

For NLG, CRSA yields higher diversity and lower redundancy while maintaining comparable quality, demonstrating its ability to assess context-adaptive generation beyond templated responses. Overall, by integrating rich linguistic phenomena with business-logic constraints, CRSA provides a more discriminative benchmark that—together with its fine-grained annotations and new evaluation metrics—probes and advances the true capability boundaries of TOD models in complex scenarios.

### 4.3.2 USER SIMULATOR TRAINING WITH CRSA

User simulators are essential components in TOD system (Lin et al., 2021). To assess CRSA's suitability, We train user simulators on Qwen1.5 and Baichuan2 (1.8B–14B) using supervised context + system utterance → user response. Test dialogs include slot-value replies and chit-chat perturbations. Evaluation targets semantic quality and diversity using BLEURT, Distinct-2, Parse Tree Diversity (PTD), and Semantic Embedding Variance (SEV) (definitions in Appendix C.5).

Table 9: Performance improvements of user simulators with CRSA fine-tuning

| Model | BLEURT | Distinct-2 | PTD | SEV |
|---|---|---|---|---|
| Qwen1.5-1.8B | +6% | +10% | +16% | +12% |
| Qwen1.5-7B | +8% | +16% | +19% | +21% |
| Qwen1.5-14B | +4% | +14% | +13% | +17% |
| Baichuan2-7B | +7% | +9% | +14% | +14% |
| Baichuan2-13B | +4% | +15% | +21% | +12% |

As summarized in Table 9, CRSA fine-tuning yields consistent and statistically significant gains across all model sizes. Improvements are most pronounced for medium-sized models, indicating better structural variety, higher semantic variability, stronger contextual adaptation, and reduced repetition. These findings show that CRSA provides an effective training corpus for user simulators, improving both behavioral realism and generative diversity.

## 5 CONCLUSION

This paper introduces CRSA, the largest Chinese TOD dataset targeting real-world business scenarios. It provides multi-dimensional, fine-grained semantic annotations with high-fidelity coverage of practical service contexts, and emphasizes system-led control and exception handling to strengthen process modeling and controllable response generation. Its hierarchical annotation framework offers rich semantic and strategic supervision. Experiments demonstrate that CRSA enhances semantic depth, pragmatic complexity, and training utility. Benchmark evaluations reveal its ability to expose model performance boundaries, while user simulator experiments confirm its support for realistic and diverse behaviors. These findings position CRSA as a challenging and valuable resource for TOD research and high-quality dialogue system development.

## ETHICS STATEMENT

We adhere to the ICLR Code of Ethics and commit to responsible stewardship of research. Our study focuses on task-oriented dialogue (TOD) modeling and controllable generation, and we have taken steps to minimize risks to individuals and society while promoting transparency, reproducibility, and fairness.

**Human subjects and privacy.** This work does not involve intervention with, or collection of personally identifiable information (PII) from, human subjects. Data used for training and evaluation were curated from publicly available or appropriately licensed resources and/or synthetically generated for research purposes. All examples were de-identified to prevent re-identification. Annotators (where applicable) were informed about the research purpose, instructed to avoid sensitive content, and provided consent prior to annotation.

**Licensing and data sharing.** We respect original licenses and redistribution terms. Any artifacts we release (e.g., code, prompts, evaluation scripts) will comply with upstream licenses. When redistribution of third-party data is restricted, we provide scripts to reproduce the processed data from the original sources. All links provided for review are anonymized to preserve double-blind review.

**Fairness, bias, and potential harm.** Language technologies may amplify social biases or enable misuse (e.g., discrimination, manipulation, or privacy violations). We mitigate these risks by (i) avoiding sensitive attributes in supervision signals; (ii) auditing outputs for obvious stereotypes and toxicity; and (iii) providing control mechanisms (e.g., safe deviation handling) designed to reject or redirect unsafe behaviors. We encourage independent audits and responsible downstream use.

**Safety and dual use.** The methods are intended for assistive and research purposes in TOD systems. They are *not* designed for surveillance, profiling, or other applications that could harm individuals or communities. We caution against such uses and discourage deployment in high-risk settings without comprehensive safety safeguards and domain expert review.

**Scientific integrity and transparency.** We report methods and settings with sufficient detail for replication and avoid fabrication, falsification, or misleading claims. Evaluation protocols are described to enable reproducibility; code and configuration files will be made available in an anonymized repository during review and, upon acceptance, in a public repository.

**Compute and environmental considerations.** We aimed to limit environmental impact by selecting modest model sizes/compute where possible, reusing checkpoints, and prioritizing efficient training and evaluation. We encourage practitioners to consider energy and carbon costs when scaling.

REPRODUCIBILITY STATEMENT

We have taken deliberate steps to ensure that our work is reproducible. The dataset construction, annotation pipeline, and processing steps are described in detail in Section 3 (*Dataset*) of the main paper, while additional specifications on data formats, normalization, and annotation details are provided in Appendix E. To facilitate independent verification, we release an anonymized repository at `https://anonymous.4open.science/r/CRSA-CBBB`, which contains the full CRSA dataset, preprocessing scripts, trained models, and experimental code with documentation. This repository also includes instructions for reproducing all experiments reported in the paper. For model design, training objectives, and evaluation protocols, we provide descriptions in Sections 3.3 and 4. Hyperparameters, ablation settings, and implementation details are reported in Sections 4. Together, these resources ensure that both our data and experimental results can be independently replicated, thereby supporting transparency, verification, and future extension of our work.

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

# APPENDIX

## USE OF LLMS

In this work, large language models (LLMs) were used in a limited and controlled manner. Specifically, during the dataset collection stage, LLMs were employed to generate a small portion of dialogue data (less than 30% of the final corpus) as described in Section 3.1. These automatically generated samples were not directly included in the dataset; instead, they underwent two rounds of manual verification, modification, and filtering before being integrated into the final CRSA dataset to ensure quality and reliability. For the writing of this paper, LLMs were only used for English grammar checking and spell correction. They did not contribute to the research design, experimental results, or substantive content of the paper. All methodological contributions, analysis, and interpretations remain the responsibility of the authors.

# A   DEFINITIONS OF KEY CONCEPTS

This appendix provides formal definitions of several central concepts used throughout the paper, including *complex business scenarios*, *semantic diversity*, and *process control*. These definitions supplement the main discussion and clarify the evaluation perspectives underlying the CRSA dataset.

## A.1   DYNAMIC SCENARIOS

Throughout the paper, we use the term *dynamic scenarios* to describe the intra-domain dynamism inherent in real-world business dialogues. To avoid potential ambiguity with multi-domain switching or complex coreference phenomena, we define this term strictly within a single business domain as follows:

- **Evolving user goals and constraints:** User objectives are incrementally revealed across turns and are frequently revised, supplemented, or overridden during the interaction.
- **Preference emergence and fluctuation:** User preferences (e.g., price tolerance, timing flexibility) surface gradually through candidate comparison and may shift multiple times.
- **Non-linear dialogue structures:** Conversations may involve backtracking, stage jumping, subtask insertion, and interruption–resumption patterns.
- **Process-level adjustments:** The system must actively re-align stages, restart subprocesses, or re-clarify key slots to maintain dialogue-state and transaction consistency under evolving user inputs.

In the revised manuscript, we avoid the broad term "dynamic scenarios" and instead adopt the more precise expression: *"dialogue settings where user goals, constraints, and preferences continuously evolve and undergo multiple revisions within multi-turn interactions."* This definition is supported through quantitative statistics (e.g., modification frequencies, goal-change ratios) and representative cases presented in the dataset.

## A.2   SYSTEM-DRIVEN DIALOGUE

We further clarify the term *system-driven dialogue*, which in this work refers to a specific, operationally defined interaction paradigm rather than the general notion of "proactive systems." The definition is grounded in both process structure and behavioral design:

**(1) Process-level control.** The dialogue is organized into three explicit stages—information acquisition, candidate selection and comparison, and completion/confirmation. Stage progression is primarily triggered by the system based on dialogue-state tracking and task-progress signals, rather than passively following user prompts. To prevent conversation stalls, the system employs richer response strategies such as proactive clarification, recommendation, and adaptive style switching.

**(2) Behavior-level structure.** The system behavior space consists of 63 predefined actions, each mapped to a unique stage, forming a structured strategy space defined by the Cartesian product of *(stage × behavior type)*. Additionally, four dimensions of controllable personalization labels (interaction style, deviation-handling strategy, task-irrelevant question handling, and process compactness) specify stylistic realizations of the same underlying strategy.

Thus, we define *system-driven dialogue* as: *"a dialogue setting in which the system assumes primary responsibility for process advancement—both in stage transitions and system-action selection—supported by explicit stage annotations, structured behavior triplets, and controllable behavior-style labels."*

## A.3 COMPLEX BUSINESS SCENARIOS

In this work, *complex business scenarios* refer to task environments characterized by multi-factor constraints, evolving objectives, and high interaction variability. Such scenarios typically exhibit the following properties:

- **Multi-slot, multi-constraint, multi-goal interactions:** Tasks involve heterogeneous slot types (e.g., time, price, eligibility, route constraints) with strong cross-slot coupling.
- **Incremental and under-specified user needs:** User goals are often revealed gradually across multiple turns rather than provided in a single query.
- **Context dependence and preference evolution:** Task progress requires synthesizing dispersed contextual cues, while user preferences (e.g., price tolerance, scheduling preference) evolve with presented options.
- **Frequent non-canonical behaviors:** Users regularly exhibit off-topic turns, revisions, goal changes, or subtask insertions, introducing irregularities uncommon in standard datasets.
- **Real transaction and compliance constraints:** Decisions involve monetary calculations, fare differences, business rules, and user-sensitive constraints.

Airline booking is a representative instance of such scenarios, combining high financial stakes, strict operational rules, multi-turn option comparison, and inherently fuzzy user preferences.

## A.4 SEMANTIC DIVERSITY

We define *semantic diversity* as a multidimensional property capturing the breadth and variability of linguistic and task semantics within a dialogue dataset. To ground the term in quantifiable aspects, we adopt the following measurable dimensions:

**User expression level.**

- Higher proportion of fuzzy or subjective expressions (e.g., "as cheap as possible", "not too early").
- Greater frequency and variety of anomaly behaviors (e.g., ambiguous turns, conflicting constraints, off-topic utterances).

**Slot and intent level.**

- Larger slot-value space and a higher proportion of subjective or bounded-slot values.
- Increased prevalence of multi-turn revisions, corrections, and constraint conflicts.

**System behavior and response level.**

- Broader distribution of system behavior types beyond fixed templates.
- Higher Distinct-n and lower Self-BLEU scores indicating less templated, more varied language generation.

We emphasize that lower DST/DA scores alone do not directly imply higher semantic diversity. Rather, we interpret these performance differences jointly with the above statistics and downstream model behaviors, collectively evidencing the increased semantic and pragmatic challenges posed by CRSA.

### A.5   PROCESS CONTROL

*Process control* refers to the model's ability to select appropriate system actions and dialogue stages across multi-turn interactions such that the task can be reliably completed within a limited number of turns. Formally, process control encompasses:

- **Stage reasoning:** Correctly identifying the current dialogue stage and determining when to advance or remain within a stage.
- **Strategy selection:** Producing system behaviors aligned with task progression (e.g., inquiry, confirmation, comparison, decision-making).
- **Exception handling:** Maintaining task correctness under noisy, ambiguous, or inconsistent user inputs.

In our experiments, we quantify process control using structural metrics such as ADFC, TCR, STA, and end-of-dialogue timing accuracy. Compared with existing datasets that lack explicit stage labels and system-action supervision, CRSA provides structured signals that directly support the learning of process-aligned dialogue strategies.

## B   SYSTEM AND USER GUIDELINES FOR CROWDSOURCED DIALOGUE SIMULATION

To ensure consistency, controllability, and naturalness in crowdsourced dialogue generation, we design two structured protocol documents: **System-side Dialogue Behavior and Response Strategy Specification** and **User-side Requirement Expression and Interaction Process Protocol**. This appendix provides a systematic and detailed introduction to these two specifications.

### B.1   SYSTEM-SIDE DIALOGUE BEHAVIOR AND RESPONSE STRATEGY SPECIFICATION

This document is designed for crowd workers playing the role of the system, guiding system behavior with consistent task progression and clear strategic responses.

#### B.1.1   DIALOGUE PHASES AND SLOT INTERACTION RULES

The system dialogue process is divided into three main phases:

- **Phase 1: Basic Information Collection**
  Required slots: *departure city, destination, departure time, personal information (name, phone)*
  Objective: Initiate the dialogue and gather essential booking information.
- **Phase 2: Candidate Option Recommendation and Selection**
  Optional slots: *price, airline, cabin class, transfer option, airports, flight duration*
  Objective: Provide candidate flight options and guide user selection or preferences.
- **Phase 3: Supplementary Details and Task Completion**
  Optional slots: *seat preference, meal inclusion, luggage allowance, discount policies*
  Objective: Confirm auxiliary services, summarize booking status, collect identity info, and close dialogue.

For required slots, the system must use explicit inquiries and confirmation strategies. For optional slots, the system uses soft questioning, default filling, or conditional guidance.

### B.1.2 DIALOGUE FLOW CONTROL AND BEHAVIOR DECISION RULES

The system's next action in each round is decided based on user replies:

- If **user replies as expected**: continue querying unfilled slots or move to the next phase.
- If **user replies unexpectedly**:
  - If the reply matches one of the six predefined user anomaly types (refusal, counter-question, repetition, irrelevance, vagueness, aggression), select the proper recovery action from the *User Anomaly Handling Guide*.
  - If not, initiate repair strategies (e.g., clarification, confirmation, guided redirection).
- If **user reply is ambiguous**: enter the "problem repetition" procedure (rephrasing, clarification, scope narrowing).

### B.1.3 USER QUESTION CLASSIFICATION AND SYSTEM REPLY POLICY

User questions are categorized as:

- **Slot-related questions**: provide complete and precise answers.
- **Business-related but slot-irrelevant questions**: answer informatively to aid user decision-making.
- **Task-irrelevant questions**: respond briefly and reroute dialogue using predefined fallback strategies.

### B.1.4 SYSTEM DIALOGUE ACTION INVENTORY

We define 63 types of system actions, including:

*greeting, slot inquiry, confirmation, clarification, recommendation, summary, rejection, redirect, reminder, database query, context recall, state transition.*

Each action type includes usage conditions and example utterances to assist crowd workers in response construction.

### B.1.5 SYSTEM-SIDE KNOWLEDGE SUPPORT

The document provides all necessary knowledge for system response:

- Slot dependencies,
- Required personal fields,
- Booking domain knowledge,
- Reasonable inference rules,
- General world knowledge.

All resources are embedded as rules or natural text for in-task reference.

### B.2 USER-SIDE REQUIREMENT EXPRESSION AND INTERACTION PROCESS PROTOCOL

This document is provided to crowd workers simulating the user role. It encourages flexible, personalized, and realistic expression aligned with business goals and task dynamics.

### B.2.1 EXPRESSIVE SCOPE AND TASK GOAL CONSTRUCTION

Users are allowed to interact over 15 slots, including required and subjective/ambiguous ones. They are guided to:

- Set realistic booking goals,
- Proactively express needs during the dialogue,
- Dynamically modify preferences, constraints, and query directions.

### B.2.2 EXPRESSION FLEXIBILITY AND INTERACTION STRATEGIES

Users are encouraged to produce natural, diverse, and non-linear dialogues:

- **Linguistic diversity**: informal, vague, or elliptical expressions,
- **Behavioral dynamics**: interruptions, goal changes, nested reasoning,
- **Tactical misalignment**: posing unexpected or strategic questions to test system robustness.

### B.2.3 BACKGROUND KNOWLEDGE AND TASK SUPPORT

To support realistic user construction, the protocol provides domain-specific knowledge (slot definitions, booking logic, etc.) and interaction strategies.

**Examples include:**

- Goal revision across dialogue turns,
- Contextual chit-chat insertions,
- Ambiguous preferences,
- Adversarial moves or deviations from task flow.

These two protocol documents ensure a structured and semantically rich framework for large-scale, high-quality dialogue simulation, enabling fine-grained control over both system-led task execution and user-side diversity. They serve as foundational design components behind the CRSA dataset and its empirical robustness.

## B.3 QUALITY CONTROL FOR LLM-AUGMENTED DIALOGUES

*Note: The LLM-generated subset of CRSA (20.3%) was retained only after a structured multi-stage validation pipeline combining prompt constraints, automated logical checks, and human auditing to ensure consistency with real dialogue distributions.*

To ensure that LLM-generated dialogues matched the linguistic and behavioral realism of human-produced interactions, a three-step quality control protocol was applied.

### B.3.1 STRUCTURED PROMPT DESIGN

Generation was guided by domain-informed prompts with few-shot examples. Each dialogue was required to: (i) follow a three-stage WOZ process (information gathering → candidate comparison → confirmation/transaction) with system-led progression; (ii) include one or more natural deviation behaviors (e.g., ambiguity, revision, subgoal insertion) and recovery strategies; and (iii) disclose user intent incrementally, while maintaining realistic tone, pacing, and information density.

### B.3.2 AUTOMATED STRUCTURAL AND LOGICAL SCREENING

Generated dialogues were automatically evaluated using an LLM reviewer to check: turn alternation consistency, phase order validity, slot coherence (including prices and constraints), and alignment between annotated anomalies and recovery behaviors. Clearly invalid samples were discarded; borderline cases were flagged for manual inspection.

### B.3.3 HUMAN REVIEW AND DISTRIBUTION ALIGNMENT

Dialogs that passed automated screening were manually reviewed in batches for pragmatic naturalness, domain appropriateness, and business rule consistency. Reviewers also ensured that anomaly frequency, dialogue length, and behavioral patterns aligned with the statistical distributions of real and crowdsourced portions, preventing systematic stylistic drift from synthetic generation.

After this process, only dialogues demonstrating structural coherence, realistic error patterns, and natural conversational flow were retained. Additional examples and validation traces are provided in the supplementary materials for transparency.

## C EVALUATION METRICS USED IN EXPERIMENTS

This appendix provides a comprehensive description of the evaluation metrics employed throughout our experiments. Metrics are categorized into five groups based on their application in different experimental settings.

### C.1 STANDARD METRICS FOR TOD TASKS

The following metrics are widely adopted in evaluating task-oriented dialogue systems:

- **DST Accuracy (DST Acc.)**: Accuracy of predicted dialogue state (slot-value pairs) against ground truth per turn.
- **API Accuracy (API Acc.)**: Accuracy of API call decisions made by the system, including correctness of parameters.
- **Dialogue Act Accuracy (DA Acc.)**: Accuracy of dialogue act classification at each turn.
- **Dialogue Policy Accuracy (DP Acc.)**: Accuracy of predicted system action types or strategy labels.
- **Response Generation Accuracy (RG Acc.)**: Accuracy of the generated system response compared to reference.
- **Intent Accuracy**: Accuracy of predicted user intent in NLU tasks.
- **Action F1**: Macro F1 score of system action prediction in dialogue policy modeling.
- **Slot F1**: Macro F1 score of slot prediction in DST.
- **Joint Goal Accuracy**: Proportion of turns where all slots in the dialogue state are correctly predicted.
- **BLEU**: Measures n-gram overlap between generated and reference responses.
- **ROUGE-L**: Measures longest common subsequence similarity between generated and reference text.
- **Distinct-2 (%)**: Ratio of unique bigrams to total bigrams in generated responses, reflecting diversity.

### C.2 METRICS FOR ABLATION STUDY

**Stage Transition Accuracy (STA).** The STA metric evaluates the model's ability to identify the correct stage in multi-phase task-oriented dialogues. For effective dialogue management, a system must accurately recognize the current task stage based on the dialogue context. STA is calculated by comparing the model-predicted stage with the human-annotated gold label at each turn:

$$\text{STA} = \frac{1}{N} \sum_{i=1}^{N} \mathbb{⊮}(p_i^{\text{pred}} = p_i^{\text{gold}})$$

where $p_i^{\text{pred}}$ denotes the predicted stage at the $i$-th turn, $p_i^{\text{gold}}$ is the gold label, $N$ is the total number of system turns, and $\mathbb{⊮}$ is the indicator function. Higher STA indicates better recognition of dialogue progression and more precise process control.

**End-of-Dialogue Recognition Accuracy (EDR).** EDR measures the model's ability to determine whether the dialogue should be terminated. In task-oriented settings, recognizing whether all user needs have been met is essential for avoiding redundant turns and improving interaction efficiency. The metric is defined as:

$$\text{EDR} = \frac{1}{M} \sum_{j=1}^{M} \mathbb{1}(y_j^{\text{pred}} = y_j^{\text{gold}})$$

where $y_j^{\text{pred}}$ indicates whether the system generated a termination signal at the $j$-th turn, $y_j^{\text{gold}}$ is the annotated ground truth, and $M$ is the number of turns with potential end-of-dialogue conditions. EDR reflects the model's understanding of task completion and its ability to conclude the dialogue appropriately.

**Task Completion Rate (TCR).** TCR is one of the core metrics for task-oriented dialogue evaluation. It measures whether the system has successfully fulfilled all explicit user requests. For each dialogue, the user's goal slots are extracted and compared with the set of slot values fulfilled by the system:

$$\text{TCR} = \frac{1}{K} \sum_{k=1}^{K} \mathbb{1}(G_k \subseteq S_k)$$

where $G_k$ is the set of goal slots in the $k$-th dialogue (annotated or inferred), $S_k$ is the set of slots fulfilled by the system, and $K$ is the total number of test dialogues. A dialogue is considered successful only if all goal slots are fulfilled ($G_k \subseteq S_k$). TCR reflects the system's overall task effectiveness.

## C.3 METRICS FOR DIALOGUE SYSTEM'S ABILITY EVALUATION

### C.3.1 ADFC: AUTOMATED DIALOGUE FLOW CONTROL

**Overview of ADFC Metric** To quantitatively evaluate the dialogue system's ability to control task flow and manage dialogue progression, we propose ADFC , a composite metric that assesses both the rationality of phase transitions and the completeness of slot acquisition across dialogue turns. This metric reflects the system's capacity to guide multi-turn interactions in accordance with domain-specific task progression requirements.

The ADFC score is computed as a weighted sum of two components: **Stage Transition Score ($T_{\text{score}}$)** and **Slot Completeness Score ($S_{\text{score}}$)**, where the weights $\alpha$ and $\beta$ control the relative importance of structural flow and slot accuracy:

$$\text{ADFC} = \alpha \cdot T_{\text{score}} + \beta \cdot S_{\text{score}} \quad (\alpha + \beta = 1)$$

Following empirical studies, we set $\alpha = 0.6$, $\beta = 0.4$ to reflect a slight emphasis on task-phase correctness.

**Stage Transition Score ($T_{\text{score}}$)** This component measures how closely the model-driven dialogue stage transitions follow domain-appropriate task flows. Formally, we define::

$$T_{\text{score}} = 1 - \frac{1}{n^2} \sum_{i=1}^{n} \sum_{j=1}^{n} |M(i,j) - O(i,j)|$$

Where

- $n$ is the number of defined dialogue stages.

- $M(i, j)$ is the ideal transition matrix, which encodes the expected stage transition probabilities, estimated from the training data's business logic flow. To ensure its reliability, $M$ was verified on an independent validation set not used in model training, confirming its empirical consistency.
- $O(i, j)$ is the observed frequency of transitions between stage $i$ and $j$ in model-generated responses over the test set.

A higher $T_{score}$ indicates that the system's stage navigation aligns well with the logical business progression, enabling smooth and controllable dialogue flow.

**Slot Completeness Score ($S_{score}$)**  This score assesses whether the system collects required slot values efficiently, especially in earlier stages of the conversation. The formula incorporates a time-decay function to prioritize early-slot acquisition:

$$S_{score} = \sum_{t=1}^{T} \left( \frac{C_t + 0.5P_t}{S_{p_t}} \cdot e^{-\lambda t} \right)$$

Where

- $T$ is the total number of dialogue turns.
- $S_{p_t}$ is the number of required slots in the current phase $p_t$.
- $C_t$ is the number of correctly filled required slots at turn $t$.
- $P_t$ is the number of partially filled slots(e.g., vague expressions needing confirmation).
- $\lambda$ is the temporal decay coefficient, controlling how much emphasis is placed on early information capture.

In our experiments, we selected $\lambda = 0.1$, following an ablation-based tuning process: models were evaluated on a development set across multiple $\lambda$ values, and 0.1 yielded the best correlation with manual assessments of dialogue efficiency. This value strikes a balance between penalizing delayed information acquisition and avoiding instability from over-weighting the first few turns.

**Evaluation Capability and Suitability**  As a composite metric designed to evaluate dialogue flow control in task-oriented dialogue systems, ADFC captures both structural progression and semantic slot completion. It offers a balanced assessment of how well the system advances through task stages and gathers essential information. With its capability for large-scale automated computation, ADFC is particularly suitable for evaluating complex multi-turn dialogues. Compared to conventional metrics such as F1 or BLEU, which primarily focus on surface-level linguistic quality, ADFC better aligns with task-driven performance goals and provides more informative diagnostic signals for system-level optimization.

### C.3.2 CRAM: CONTEXTUAL RESPONSE APPROPRIATENESS METRIC

**Scoring Structure and Formula**  CRAM is a comprehensive human evaluation metric for assessing the contextual relevance of dialogue responses. Its design is grounded in Grice's maxims - particularly *coherence, relevance, and cooperativeness* - as well as the logic for the completion of tasks-oriented dialogues.

The metric comprises the following dimensions:

- **Coherence (C)** $\in [0, 3]$: Whether the response logically follows the previous context.
- **Resolution (R)** $\in [0, 2]$: Whether the response fulfills the user's request.
- **Proactivity (P)** $\in [0, 1]$: Whether the system proactively drives the task forward.

The final score is computed as:

$$\text{CRAM} = \frac{C + R + P}{6} \times 100\%$$

The final score ranges from 0% to 100%. A higher score indicates a more appropriate response and smoother task progression.

**Quality Control Protocol   Evaluator Training**: All evaluators are trained using 20 curated examples (including both positive and negative cases). Inter-rater reliability is measured by Krippendorff's $\alpha$, which must exceed 0.8.

**Triple-Blind Review**: Each dialogue is independently scored by three annotators. The final result is computed as the median score.

**Stratified Sampling**: Dialogues are sampled to cover a balanced distribution of task complexity:

- Simple tasks (single intent, $\leq 3$ turns): 30%
- Composite tasks (nested intents, 4–6 turns): 50%
- Abnormal tasks (conflicting/ambiguous intents): 20%

**Application and Validity   Discriminative Power**: CRAM successfully differentiates rule-based and neural models .

**Correlation Validation**: CRAM shows a strong correlation with human satisfaction scores.

**Diagnostic Utility**:

- **Low C Score:** Poor context management, often due to state tracking failures.
- **Low R Score:** Indicates misunderstanding or retrieval errors.
- **Low P Score:** Suggests lack of strategy or task initiative.

## C.4   METRICS FOR CONTROLLABILITY AND FLOW AWARENESS

**Automatic Evaluation**:

- **Slot Compliance Rate (SCR)**:SCR evaluates whether the slot targeted in a system-generated question aligns with the expected slot set for the current dialogue stage. It measures the system's ability to understand the current task phase and generate appropriate slot-seeking behavior.

$$\text{SCR} = \frac{1}{N} \sum_{i=1}^{N} \mathbb{K}(q_i \in S_{p_i})$$

  Where:
  - $q_i$ is the slot involved in the system-generated question at turn $i$;
  - $S_{p_i}$ is the standard slot set associated with stage $p_i$;
  - $N$ is the total number of system question turns in the test set.

- **Process Advancement Effectiveness (PAE)**:PAE measures whether the system adopts effective dialogue advancement strategies appropriate to the current task context. We define six categories of advancement templates (see Appendix E) and check whether the generated behavior matches one allowed by the contextual state.

$$\text{PAE} = \frac{1}{T} \sum_{t=1}^{T} \mathbb{K}(s_t \in \mathcal{A}(h_t))$$

  Where:
  - $T$ is the number of total dialogue turns;
  - $s_t$ is the system's behavior type at turn $t$;
  - $\mathcal{A}(h_t)$ is the set of acceptable advancement actions given context $h_t$.

**Human Evaluation (Controllability)**:

- **Style Control Consistency (SCC)**: Whether response matches desired style token.
- **Deviation Handling Consistency (DHC)**: Whether system executes designated correction strategy.

Table 10: Examples of slot value semantic normalization in CRSA.

| Slot name | User expression (raw) | Normalized form |
|---|---|---|
| Departure time | "Early morning flight" / "Around 1pm in the afternoon" | 06:00–09:00 / 13:00 |
| Destination city | "Magic City" / "Capital Airport" | Shanghai / Beijing |
| Seat class | "The more expensive, the better" / "high-end" / "with discounts" | business / first / economy |
| Price range | "cheaper" | less than 1000 |
| Date | "Before the end of the month" | before 2025-xx-31 |

- **Query Response Strategy Consistency (QRSC)**: Whether information reply matches expected control tag.
- **Flow Control Consistency (FCC)**: Whether system follows token-defined stage order and slot query sequence.

All human-evaluated metrics use expert blind ratings with reference guidelines and structured rating templates.

### C.5 METRICS FOR USER SIMULATOR EVALUATION

To evaluate the linguistic quality and diversity of generated user utterances, five automated metrics are employed:

- **BLEURT**: Measures semantic similarity between generated and reference user utterances using the BLEURT-base-zh model. Captures paraphrasing and syntactic variation.
- **Distinct-2**: Proportion of unique bigrams in generated utterances. Higher values indicate greater language variation:

$$\text{Distinct-2} = \frac{|\text{Unique Bigrams}|}{|\text{Total Bigrams}|}$$

- **Parse Tree Diversity (PTD)**: Measures structural variation using syntactic dependency parsing:

$$\text{PTD} = \frac{|\text{Unique Dependency Trees}|}{|\text{Total Sentences}|}$$

- **Semantic Embedding Variance (SEV)**: Assesses semantic diversity via Sentence-BERT embeddings:

$$\text{SEV} = \text{Mean}\left(\text{EigenValues}\left(\text{Cov}(v_1, ..., v_n)\right)\right)$$

- **Self-BLEU**: Measures intra-set similarity among generated utterances. Lower values imply higher lexical diversity:

$$\text{Self-BLEU} = \frac{1}{N}\sum_{i=1}^{N}\text{BLEU}(s_i, S_{-i})$$

## D SLOT VALUE NORMALIZATION AND SUBJECTIVE SLOT MAPPING

### D.1 SLOT VALUE SEMANTIC NORMALIZATION TABLE

To address diverse, ambiguous, and non-standard user expressions in real-world dialogues, we construct a slot value semantic normalization table tailored to the airline ticket booking domain. This table maps natural language expressions to structured, machine-readable slot categories or numerical ranges, facilitating robust slot extraction and reasoning.

The normalization process supports both hard constraint resolution (e.g., filtering flight candidates) and soft preference learning (e.g., ranking suggestions). All mappings are derived via domain-informed rule templates and verified through annotation consistency audits.

## D.2 SUBJECTIVE SLOT MAPPING STRATEGY

In realistic service-oriented dialogue, many user expressions reflect soft preferences or subjective needs that cannot be directly mapped to deterministic slot values. To bridge this gap, CRSA introduces a subjective slot interpretation mechanism based on intermediate semantic tags and context-aware resolution.

We define a set of intermediate subjective intents (e.g., "cheap", "fast", "safe", "flexible") and associate them with predefined slot configurations or value constraints.

**Example 1: Preference for Safety**

- User input: "I want to take a bigger and more reliable airline"
- Mapped intent: `airline_preference = major_carrier`
- Constraint: Restrict candidate flights to `[Air China, China Eastern, China Southern]`

**Example 2: Temporal Flexibility**

- User input: "Any departure time is fine, as long as it's cheap"
- Mapped intents: `time_flexibility = high, price_preference = low`
- Constraint: Expand time window, prioritize lowest fare options

**Example 3: Comfort-Oriented Request**

- User input: "I don't want it to be too crowded, a slightly more comfortable seat"
- Mapped intent: `seat_comfort = medium_or_high`
- Constraint: Exclude "economy basic"; prioritize "premium economy", "business"

**Mapping Workflow**  The subjective slot mapping strategy follows a three-step process:

1. **Expression Extraction:** Identify subjective expressions using keyword and pattern matching.
2. **Intent Mapping:** Map expressions to predefined intermediate intent labels via lexicon-based matching and light-weight classifier support.
3. **Constraint Resolution:** Apply intent-dependent constraints to system response candidates (e.g., candidate filtering, reranking).

# E  CRSA ANNOTATION SCHEMA AND FULL EXAMPLE

To support multi-granularity modeling of task reasoning, preference evolution, anomaly handling, and controllable system behavior, CRSA employs a stage-centric hierarchical annotation framework. In contrast to traditional utterance-level annotation, CRSA anchors annotation at the **task stage level**, preserving the full semantic context within the active stage. This enables modeling of *process control, conditional intent, fuzzy preference resolution, and context-dependent system policy planning*.

## E.1  FULL ANNOTATED EXAMPLE

### E.1.1  EXCERPTED DIALOGUE (STAGE: INFORMATION COLLECTION, LATE SEGMENT)

*User:* "Okay... fine, let's focus on Guangzhou then. But listen—if flights tomorrow afternoon are super cheap, I **might** take half a day off and leave earlier. And—uh—no layovers, okay? Also, my colleague mentioned something about a 'Fly-Flex unlimited pass' from China Southern—do you support that discount?"

*System:* "The Fly-Flex program is typically valid only for standard economy tickets, not Premium Economy or above. Since you prefer Premium Economy and require nonstop flights, applicability may be limited. Before retrieving flight options, could you confirm whether your acceptable departure range should be **13:00–20:00**?"

### E.1.2 JSON ANNOTATION

```json
{
  "metadata": {
    "dialogue_id": "CRSA_1217",
    "segment_id": "P1_Late_06",
    "source": "real",
    "schema_version": "CRSA-v2.0"
  },

  "context": {
    "stage_name": "information_collection",
    "stage_status": {
      "is_current_stage": true,
      "stage_index": 1,
      "progress_state": "late-stage"
    },

    "stage_memory": {
      "history_utterances": [
        "User: Can I book a flight to Foshan?",
        "System: May I confirm your departure city?",
        "User: Beijing... or Shijiazhuang... or Zhangjiakou?",
        "System: Foshan has no civil airport; route to Guangzhou.",
        "User: Will flights get delayed?",
        "System: I will prioritize punctual routes.",
        "User: Tomorrow afternoon but not too late.",
        "System: Should I check 18:00{20:00 from SJW or ZQZ?",
        "User: If cheap, maybe earlier. No layovers. Fly-Flex?"
      ]
    },

    "semantic_state": {
      "structured_slots": {
        "departure_city_candidates": ["Beijing", "Shijiazhuang", "Zhangjiakou"],
        "destination": "Guangzhou",
        "flight_type": "nonstop"
      },

      "subjective_slots": {
        "departure_time_preference": {
          "raw_expression": "If cheap, I might leave earlier.",
          "normalized": {
            "preference_type": "conditional_soft_constraint",
            "baseline_window": "18:00{20:00",
            "expanded_window": "13:00{20:00",
            "trigger": "lower_price"
          }
        }
      },

      "dialogue_dynamics": {
        "preference_evolution": [
          {
            "slot": "departure_time",
            "shift_type": "soft expansion",
            "reason": "price sensitivity revealed"
          }
        ],
```

```
            "intent_clarity_state": "partially_resolved"
        }
    },

    "anomaly_tracking": {
      "has_anomaly": true,
      "anomaly_type": ["policy-irrelevant-query"],
      "trigger": "User asks coupon eligibility before selection",
      "system_recovery_strategy": "brief-answer → redirect"
    }
  },

  "dialogue": {
    "current_exchange": {
      "user_utterance":
        "If flights are cheap I may leave earlier. No layovers. Fly-Flex?",
      "system_response":
        "Fly-Flex may not apply. Should I search 13:00{20:00?"
    },

    "system_behavior_labeling": {
      "dialogue_act": ["clarify", "confirm", "filter"],
      "behavior_triplet": {
        "action": "refine_time_range",
        "target": "departure_time",
        "strategy": "guided-filtering"
      },
      "control_labels": {
        "style": "friendly-professional",
        "flow_control": "guided",
        "deviation_handling": "acknowledge-and-redirect",
        "irrelevant_query_policy": "brief-answer"
      }
    }
  },

  "output": {
    "system_dialogue_act": "clarify_time_range",
    "utterances":
      "Understood | Fly-Flex may only apply to standard economy. "
      "Before I pull options, should I search from 13:00 to 20:00?",
    "control": {
      "style": "warm",
      "query": "guided",
      "deviation": "redirect",
      "flow": "continue-current-stage"
    }
  }
}
```

### E.2 SUMMARY

This example demonstrates CRSA's capacity to encode: (1) persistent stage-aware reasoning, (2) structured and fuzzy slot values, (3) anomaly and recovery strategy labeling, and (4) controllable policy metadata.

Such structured annotation enables supervised training, explainable planning, and controllable response generation in complex real-world TOD settings.

## F CROSS-DOMAIN EVALUATION OF THE CRSA ANNOTATION FRAMEWORK

To further examine the generality of the proposed annotation methodology, we conduct cross-domain experiments on four heterogeneous TOD datasets: **SGD**, **CrossWOZ**, **PRESTO**, and **X-**

**RiSAWOZ**. For each dataset, we sample 200 dialogues, manually re-annotate them using the CRSA schema to establish gold references, and evaluate an automatic annotation model under zero-shot and few-shot settings. Results reveal two distinct performance groups, highlighting both the strengths and current limitations of CRSA-based annotation transfer.

**(1) Robust transfer to structurally compatible TOD datasets.** On **SGD** and **CrossWOZ**, which share similar dialogue structures and slot semantics with typical task-oriented settings, the model demonstrates strong zero-shot performance:

- Key-slot accuracy: **82%**
- Revision rate (manual corrections required): **19%**

With only two in-context demonstrations, performance further improves:

- Key-slot accuracy: **91.5%**
- Revision rate: **13%**

These results indicate that the core CRSA annotation components—such as stage transitions, preference clarification, anomaly recognition, and candidate-driven goal adjustment—are largely domain-agnostic, enabling efficient transfer across standard TOD tasks.

**(2) Sensitivity to multilingual and stylistically unconstrained corpora.** On **PRESTO** and **X-RiSAWOZ**, which contain extensive multilingual mixing, free-form discourse, dialectal markers, and implicit referential expressions, zero-shot performance drops substantially:

- Key-slot accuracy: **54.5%**
- Revision rate: **57.75%**

With two-shot adaptation:

- Key-slot accuracy: **72.75%**
- Revision rate: **36%**

The performance gap reflects the difficulty of slot normalization, anomaly detection, and stage identification when linguistic variability is high. The results also suggest that minimal in-domain demonstrations or lightweight fine-tuning can substantially improve adaptation.

**(3) Implications for the generality of CRSA.** Across all four datasets, the experiments support three conclusions:

- The structural design of CRSA exhibits strong cross-domain applicability and can be transferred beyond airline booking.
- The automatic annotation model shows robust generalization in standard TOD domains and can rapidly adapt to new scenarios with very limited supervision.
- Multilingual and highly heterogeneous dialogue styles remain challenging, though few-shot conditioning can mitigate most issues.

These findings reinforce that CRSA constitutes a reusable *complex business dialogue annotation framework*, rather than a domain-specific schema.

**(4) Ongoing extensions.** To further improve cross-domain robustness—particularly for multilingual and stylistically diverse data—we are conducting the following extensions:

- Scaling model capacity (40B–70B) for enhanced robustness.
- Unifying training to bilingual (Chinese–English) corpora for improved slot stability.
- Applying lightweight LoRA-based adaptation with few in-domain samples.
- Developing cross-lingual alignment and normalization rules.

These enhancements will be included in the final version and constitute promising directions for future work.

