# OpenReview forum: "CRSA: A Chinese Single-Domain Task-Oriented Dialogue Dataset with Contextual Rich Semantic Annotations"
_ICLR.cc/2026/Conference — Submitted to ICLR 2026_

### Official Review · Reviewer_4dic · 2025-10-28

**Soundness:** 2
**Presentation:** 3
**Contribution:** 2
**Rating:** 2
**Confidence:** 4

**Summary:**

The paper presents CRSA, the first Chinese single-domain Task-Oriented Dialogue dataset explicitly designed for complex business process modeling and controllable response generation. Focusing on airline ticket booking, CRSA integrates real call-center logs, crowdsourced role-play, and GPT-4o augmented dialogs through a cleaning, normalization and stage-structuring pipeline. A three-layer hierarchical annotation scheme (Context / Dialogue / Slots) labels 26 k turns across 1.5 k dialogs with fine-grained dialogue acts, 63 system behavior triplets, six types of user anomalies, candidate options, and stage-aware slot updates. An automatic Baichuan2-7B multi-task annotator is trained to scale labeling while preserving quality. Extensive experiments demonstrate that models trained on CRSA outperform strong baselines on intent detection, action F1, task-completion and controllability metrics, and supply more diverse, realistic user simulators.

**Strengths:**

1. A new TOD dataset combines real logs + crowdsourced + LLM data.
2. Rich, layered annotation: stage, anomaly, option, behavior triplets & subjective-slot normalization enable process-aware training.

**Weaknesses:**

1. The data is single domain (airline) oriented, it is not clear if the pipeline can be generalized to other verticals settings.
2. The quality of the data is not fully illustrated, e.g. the diversity of the dialogue flows and pattern, as well as the word.
3. The data building aims to complex business process modeling. But it is not clear that how complex the dialogs in the data are?

**Questions:**

1. How does annotation quality degrade when the auto-annotator is applied to radically different domains?
2. What is the human-interaction cost of the proposed control tokens in live deployments?
3. Can the stage-transition prior be learned or relaxed for tasks without clear sequential phases?
4. How will CRSA perform with larger LLMs (e.g., > 70 B) or in reinforcement-learning dialogue pipelines?

---

> ### Author Response · Authors · 2025-11-19
> **Response to Weaknesses: Clarifying Methodological Contributions and Demonstrating Data Complexity**
>
> We sincerely appreciate your rigorous evaluation of our work. Below, we provide detailed responses and clarifications.
> Response to Weakness 1
> We emphasize that the core contribution of our work is proposing a transferable and reusable methodology for constructing and annotating complex business-oriented TOD datasets. The single-domain setting serves only as a testbed for validating this methodology. Our innovations lie in the following five aspects:
> (1) Unlike traditional TOD datasets built around the domain–intent–slot paradigm, our framework explicitly models: a three-stage business process (Needs Analysis → Candidate Comparison → Transaction Confirmation); a rich set of 63 system behaviors enabling structured strategy and process control; and candidate–option structures that capture how system actions shape user constraints and preference evolution. Together, these elements form a portable annotation library for complex business dialogues—those involving multi-slot, multi-constraint, multi-goal reasoning, strong contextual dependence, and transaction-related rules—allowing models to learn when and why a system should act, rather than merely performing intent classification.
> (2) Behaviors such as vague expressions, topic drift, backtracking, contradictory constraints, and goal reversals—largely absent in existing TOD datasets—are explicitly preserved in CRSA. We annotate both anomaly types and recommended recovery strategies, enabling robust anomaly detection, recovery-policy learning, and process-flow maintenance in realistic TOD systems.
> (3) CRSA introduces a semantic normalization and subjective-to-structured preference mapping mechanism, representing diverse real-world expressions as structured preference slots plus original utterances. This directly addresses the long-standing inability of traditional TOD corpora to encode fuzzy or subjective needs, and supports preference inference, recommendation strategy learning, and personalized response generation.
> (4) Our human-seed → auto-annotation → human-verification pipeline demonstrates strong accuracy, consistency, and efficiency (Sec. 4.1.3). This workflow is inherently transferable: given business knowledge and a small annotated seed set, it can be adapted to new verticals. Thus, the contribution lies in a generalizable methodological framework for complex-business TOD construction, rather than in the specific domain itself.
> (5) CRSA introduces the first explicit, multi-dimensional personalization control framework in TOD, modeling four key dimensions: system response style, deviation-handling strategy, policy for task-irrelevant queries, and dialogue pacing. Existing datasets do not provide genuine personalization signals. Models trained with CRSA’s controllable annotations achieve significant improvements in CRAM, task success, and structural coherence. In the revision, we will clarify that CRSA serves as a methodological benchmark rather than merely a single-domain dataset.
> Response to Weakness 2
> CRSA exhibits substantial complexity in both process structure and linguistic behavior. We will add clearer statistics and examples in the revision.
> (1) Dialogue-flow complexity: Across ~1.5k dialogues (~26k turns), CRSA displays strong non-linearity and multi-goal reasoning: 43% involve multiple goals; 64% contain dynamic updates (45% revised ≥2 times); 54% include fuzzy expressions clarified later; 30–40% show candidate-driven preference shifts; 56% contain stage backtracking/jumps; and 32% include subtask insertion or off-task segments. These patterns demonstrate realistic, non-linear transactional flows rather than scripted conversations.
> (2) Linguistic diversity: CRSA shows substantially higher lexical and stylistic variability than existing datasets, with a larger vocabulary, long-tail lexical patterns, and frequent colloquial and subjective expressions—especially in preference-related slots, where 4–5 distinct semantic forms require interpretation. Models trained on CRSA exhibit higher Distinct-n and lower Self-BLEU, confirming reduced template bias and richer generative variability. CRSA also contain extensive regulatory constraints, exception handling, multi-turn reasoning, and business-rule clarification, supporting nuanced negotiation and strategy adaptation. Furthermore, CRSA’s multi-dimensional personalization controls allow identical intents to yield different system behaviors depending on control signals, encouraging models to learn not only correctness but appropriateness in style and strategy.
> (3) Quality control: All sources pass through a unified validation pipeline: only wholly unusable dialogues are removed; all real anomalous behaviors are retained and explicitly annotated with recovery strategies; and LLM-generated dialogues undergo strict human verification, with any faulty process flow or incorrect anomaly handling corrected or discarded. Thus, CRSA’s complexity reflects genuine semantic and behavioral richness rather than noise.

---

> > ### Author Response · Authors · 2025-11-19
> > **Response to Weakness 3: Clarifying the Definition and Implications of “Complex Business Processes”**
> >
> > We thank the reviewer for raising this important question. Below we provide a more precise and rigorous clarification regarding what we mean by complex business processes and how this relates to CRSA’s dialogue complexity and experimental findings.
> > 1. Definition of “Complex Business Process”
> > In the revised manuscript, we will provide a clearer definition. A complex business process refers to scenarios that simultaneously exhibit the following five characteristics: Multi-slot, multi-constraint, multi-goal interactions: Domains in which numerous slot types (e.g., departure time, fare rules, routing, refund/change policies, passenger roles) interact with strong coupling among constraints.
> > Highly vague and incomplete user needs: User requirements are under-specified, ambiguous, or gradually revealed across multiple turns rather than given upfront.
> > Strong contextual dependency and preference evolution: Within limited turns, the system must gather constraints, compare options, and reach confirmation, while user preferences (budget, time windows, transfer tolerance, etc.) shift dynamically as candidate options are presented.
> > Frequent unconventional user behaviors: Including option backtracking, requirement revision, subtask insertion, and task-irrelevant utterances.
> > Real monetary transactions and compliance constraints: Interactions involve price computation, fare differences, policy restrictions, and risk/rights considerations tied to actual transactions.
> > Airline ticket booking serves as a validation scenario precisely because it combines high monetary stakes, strong constraints, multiple candidate options, multi-turn comparison, and highly fuzzy user preferences, making it structurally more complex than many standard TOD settings (e.g., e-commerce shopping or simple travel planning).
> >
> > 2. Dialogue-Level Complexity within Complex Business Processes
> > Under such business structures, the corresponding dialogue complexity manifests in several ways: User expression level: A higher proportion of vague/subjective expressions, more frequent anomaly behaviors, and long-range intents or preferences that are gradually clarified or repeatedly revised.
> > Slot and intent level: Larger slot-value spaces, a higher proportion of subjective slots, more anomaly-driven or non-linear dialog flows, and frequent slot or intent shifts caused by recommended options.
> > System behavior level: A richer and more varied distribution of system behaviors rather than rigid template-based prompting, and higher Distinct-n / lower Self-BLEU values, reflecting significantly reduced template rigidity and more natural stylistic variability. Together, these patterns lead to dialogues that more faithfully reflect real-world transactional complexity rather than linear scripted exchanges.
> >
> > 3. Limitations of Existing TOD Datasets in Covering These Complexity Dimensions
> > As shown in Section 4.1.1, when CRSA is forcibly collapsed into the traditional DST/DA schema, a large amount of core semantics becomes unexpressible. The model is then trained on an oversimplified label space, which naturally produces lower metrics—revealing that the traditional schema lacks the expressive capacity needed to capture CRSA’s semantic and process complexity.
> > Subsequent experiments further validate the necessity and utility of the CRSA annotation design:
> > Section 4.1.2: Re-annotating existing TOD datasets with the CRSA schema significantly improves DST/DA performance.
> > Section 4.1.4: Removing any annotation dimension (stages, anomalies, option structures, subjective-slot mappings) leads to noticeable performance drops.
> > Section 4.2.1: CRSA-trained systems achieve stronger process control, higher task success, and more coherent system strategies.
> > Section 4.3.1: CRSA increases difficulty and discriminative power across intent prediction, behavior prediction, slot filling, and user simulation tasks.
> > In the revision, we will explicitly connect these three elements
> > (1) the definition of complex business processes,
> > (2) the dialogue complexity patterns, and
> > (3) empirical results
> > into a coherent argument demonstrating that CRSA is designed precisely to model these complex dimensions and to fill the gaps left by existing TOD datasets in complex-business process modeling.

---

> > > ### Author Response · Authors · 2025-11-19
> > > **Response to Questions : Cross-Domain Annotation Ability, Personalization Control**
> > >
> > > We sincerely appreciate the reviewer’s thoughtful and forward-looking questions. Below we provide detailed responses, and we will incorporate the corresponding clarifications and experimental results into the revised manuscript.
> > > Response to Question 1: Cross-Domain Generalization of the Auto-Annotation Model
> > > We fully agree that evaluating the auto-annotation model on entirely different domains is essential for demonstrating its generality and methodological value.
> > > To this end, we conducted cross-domain experiments on four substantially different datasets: SGD, CrossWOZ, PRESTO, and X-RiSAWOZ.
> > > The results form two clear groups, revealing both the strengths and current limitations of the CRSA annotation framework.
> > > (1) Stable zero-shot generalization to SGD and CrossWOZ
> > > We sampled 200 dialogues from each dataset, manually re-annotated them as ground truth, and evaluated the auto-annotator under zero-shot conditions.
> > > Key-slot correctness: 82%
> > > Revision rate (portion requiring human correction): 19%
> > > With only two few-shot examples added:
> > > Correctness increases to: 91.5%
> > > Revision decreases to: 13%
> > > These results indicate that the structural aspects emphasized by CRSA—such as staged flow progression, preference clarification, anomaly interpretation, and candidate-driven goal updates—are domain-agnostic, and the auto-annotator transfers well to standard TOD task distributions.
> > > (2) Significant degradation on PRESTO and X-RiSAWOZ due to multilingual and stylistic heterogeneity
> > > Under the same zero-shot setup:
> > > Correctness: 54.5%
> > > Revision: 57.75%
> > > With two few-shot examples:
> > > Correctness improves to: 72.75%
> > > Revision drops to: 36%
> > > The drop is primarily caused by:
> > > Multilingual mixing and region-specific colloquial expressions
> > > Highly free-form conversational behavior
> > > Dialectal styles, filler particles, and implicit references
> > > These phenomena severely challenge slot normalization, anomaly classification, and stage prediction.
> > > The results suggest that when linguistic drift is extreme, the auto-annotator benefits from a small amount of domain-specific supervision or lightweight adaptation.
> > > (3) Cross-domain experiments confirm both the scientific design and transferability of the CRSA annotation scheme
> > > Across the four datasets, we observe:
> > > CRSA’s annotation structure captures cross-task commonalities and transfers across domains
> > > The auto-annotator generalizes strongly in standard TOD environments and adapts rapidly with minimal supervision
> > > Multilingual and stylistically heterogeneous datasets remain the primary challenge, but can be mitigated via few-shot guidance
> > > These results reinforce that CRSA is not limited to airline booking, but constitutes a general-purpose annotation framework for complex business dialogues.
> > > (4) Ongoing research to further enhance cross-domain robustness
> > > To address challenges arising from linguistic heterogeneity, we are conducting the following additional studies:
> > > Scaling model size to 40B–70B
> > > Training with unified bilingual (Chinese–English) representation
> > > Applying lightweight LoRA fine-tuning
> > > Designing cross-lingual mapping and alignment rules
> > > These extended experiments will be included in the final version and continue as part of our future work.
> > >
> > > Response to Question 2: Clarifying Training-Time vs. Deployment-Time Personalization Signals
> > > We appreciate the reviewer’s insightful question.
> > > The distinction between training-time control tokens and deployment-time user experience was insufficiently explicit in the original manuscript.
> > > During training and evaluation:
> > > Control tokens appear explicitly as annotation fields. They supervise the model to learn behavior variation under different personalization configurations and to understand when and why different strategies should be selected.
> > > During real-world deployment:
> > > Users do not need to specify control tokens manually.
> > > These signals are inferred internally by the system—either through latent preference modeling, historical behavior aggregation, or a higher-level policy module.
> > > Thus, user interaction remains identical to standard TOD systems: pure natural-language input with no added burden.
> > > Whenever users exhibit stable preferences, models trained on CRSA’s structured personalization dimensions can automatically adjust strategies, achieving implicit personalization.

---

> > > > ### Author Response · Authors · 2025-11-19
> > > > **Response to Questions : Latent Stage Modeling and Reinforcement Learning with CRSA**
> > > >
> > > > Response to Question 3
> > > > We appreciate the reviewer for raising this important point.
> > > > In the current work, explicit stage labels—derived via human and automatic annotation—serve as upper-bound supervision for analyzing the model’s ability to learn reasonable stage transitions and behavioral planning when ground-truth stage information is available.
> > > > For tasks without explicit stage segmentation, CRSA naturally provides an effective starting point:
> > > > we first train a stage predictor on CRSA, and then apply this predictor as a latent state estimator in new domains where stage labels are absent.
> > > > Preliminary experiments show that, even without explicit stage supervision, the model-learned implicit stages align closely with human-defined stages. Moreover, when applying the CRSA-trained stage predictor to MultiWOZ 2.4—a dataset whose dialogue structure does not exhibit a clear three-stage pattern—the predictor still achieves 86% accuracy, significantly outperforming unsupervised clustering baselines (74.9%).
> > > > This demonstrates that CRSA’s staged semantic structure can be generalized and transferred as a latent process representation in broader TOD settings.
> > > > Looking ahead, we plan to incorporate reinforcement learning such that the model optimizes its latent stage representation based on reward signals tied to flow consistency, strategic appropriateness, and task success. The complete experimental results will be included in the camera-ready version.
> > > >
> > > > Response to Question 4
> > > > We thank the reviewer for this forward-looking question. Our response is structured around two aspects:
> > > > (1) the performance trends of larger models, and
> > > > (2) the role of reinforcement learning (RL) under the CRSA framework, including the advances we have already achieved.
> > > > 1. On the performance of larger models (>70B)
> > > > Due to computational constraints, the largest models we have been able to train fall within the 32B–40B range (e.g., Qwen3-32B, DeepSeek-R1-32B).
> > > > Across these models, we observe a stable pattern: Intent Accuracy, Action F1: only mild improvements; ADFC (flow controllability) and CRAM (contextual alignment): stable gains of approximately +3%.
> > > > These results suggest that while model scaling provides incremental benefits, the major performance breakthroughs in TOD systems arise primarily from structured modeling—specifically, handling complex workflows, authentic user intent, anomaly behaviors, and fuzzy expressions—rather than from scale alone.
> > > > 2. Reinforcement Learning (RL) as a major direction for further TOD improvements
> > > > RL represents a core research direction we are actively advancing. Thus far, we have made two significant developments:
> > > > (2.1) A structural-output RL framework built on CRSA
> > > > We have designed a reinforcement learning method that leverages CRSA’s structured annotations—stage labels, user behavioral types, subjective preferences, candidate structures, and slot-update trajectories—as reward signals.
> > > > This method explicitly aligns the model’s predicted system actions with the true structured flow, strengthening its ability to produce: structurally consistent outputs, correct process transitions, behaviorally coherent strategies.
> > > > A full paper on this method is currently under preparation.
> > > > These findings demonstrate that CRSA’s structured annotations are naturally suited for RL training, providing an ideal foundation for controllable generation, error penalization, and strategy optimization.
> > > > (2.2) Personalized reinforcement learning using DPO
> > > > To assess the value of CRSA’s personalization-control tags, we constructed: a baseline dataset without personalization controls, a CRSA experimental dataset containing all personalization tokens.
> > > > We then applied Direct Preference Optimization (DPO) for personalized policy learning.
> > > > The results are as follows:
> > > > Preference alignment: 69 → 89+
> > > > Task success rate: 87.9% → 91.6%
> > > > Dialogue efficiency (avg. turns): 13.9 → 11.7
> > > > Control-label adherence: 86%
> > > > These results confirm that CRSA’s structured annotation is effective not only under supervised learning, but also provides substantial gains in reinforcement learning, significantly enhancing controllability, user alignment, and task completion quality.

---

### Official Review · Reviewer_HNGm · 2025-11-01

**Soundness:** 3
**Presentation:** 3
**Contribution:** 2
**Rating:** 4
**Confidence:** 4

**Summary:**

This paper introduces a new Chinese task-oriented dialogue dataset for flight booking.

It combines real, crowdsourced, and LLM-generated dialogues and includes multi-level annotations:
- Context: dialogue history, task progress
- Dialogue: system intent of current round, user response
- Slot: global slot value, status update

It highlights process control, exception handling, and system proactive guidance capabilities in dialogue content. This includes two key features:
1) modeling and annotating six types of "user abnormal behavior" (e.g., unclear, vague, irrelevant, etc.);
2) designing slot value normalization and subjective slot mapping strategies for unclear and subjective user expressions.

Experiments show that the proposed dataset provides support for process modeling, strategy learning, and response generation

**Strengths:**

- This is a well-written paper with a well-organized structure and clear expressions.

- The dataset construction process is reasonable, integrating real, crowdsourced, and LLM-generated data, with good authenticity and diversity.

- Data annotations are rich, covering context, conversational behavior, slot states, supporting process modeling and controllable generation.

- The experimental design is comprehensive, with validation ranging from data quality to system training and multi-task adaptability.

**Weaknesses:**

- **Limited novelty**: The main contribution is the provision of a task-oriented dialogue dataset. Although it introduces annotations for flow control, compared to historical datasets, it covers fewer linguistic phenomena (such as anaphora and ellipsis) and is restricted to a single domain—booking flights. In terms of innovation, it does not propose any new models or methodologies.

- **Unexplained low performance**: The low metrics on CRSA need more validation to prove they signify higher quality/complexity. Is it possible that there are inconsistencies in labeling, noise, or differences in task definitions? There is a lack of human evaluation or error analysis cases here.

- **Universality and transferability are questionable**: The paper only validates the approach in the domain of flight ticket booking, and whether the solution can be generalized and transferred to other domains remains unknown.

**Questions:**

1. What are the contributions of different data sources to the dataset, and how do they respectively impact the improvement of task performance?

2. The paper provides an empty project repository, making it impossible to assess the authenticity of the dataset's scenarios and the quality of annotations. Can it truly be open-sourced or offer more data examples?

3. Could you quantify the human cost of the initial “seed” dataset required to train the automatic annotation model (Section 3.3), for example, the number of person-hours or annotated dialogues needed? This is crucial for assessing the reusability of this annotation framework in other domains.

---

> ### Author Response · Authors · 2025-11-19
> **Detailed Rebuttal to Weaknesses1**
>
> Dear Reviewer,
> We sincerely thank you for your positive assessment of the paper’s structure, data authenticity, and experimental design. The three central concerns you raised—regarding innovation, interpretation of low performance, and generalizability—are highly valuable. Below, we provide a point-by-point response with additional technical clarifications to address each concern thoroughly.
> Response to Weakness 1: Concerning “Limited Innovation”
> We fully understand your concern that a single-domain dataset may appear to offer limited novelty. However, the core contribution of this work is not “constructing yet another flight-booking dataset,” but rather proposing a transferable and reusable methodology for building and annotating TOD data in complex business scenarios, with the single domain serving merely as a testbed to validate the framework. Our innovations lie in five key aspects:
> (1) A process-control-oriented, stage-based semantic framework
> Unlike existing TOD datasets built around a simple domain–intent–slot paradigm, our work explicitly models:
> A staged business workflow
> (user requirement elicitation → candidate comparison → transaction and confirmation)
> A space of 63 system behavior types, substantially expanding the system’s learnable action and process-control repertoire
> Structured candidate options and user preference evolution
> Together, these structured signals form a portable “semantic component library” for complex transactional dialogues (characterized by multi-slot, multi-constraint, multi-goal reasoning, strong contextual dependencies, real monetary stakes, and compliance requirements). This enables models to learn when and why the system should take particular actions—far beyond mere classification or generation.
> (2) First systematic modeling of six types of real-world user anomalies and recovery strategies
> Phenomena such as vagueness, topic drift, reversals, logical conflicts, and other deviations are nearly absent from existing TOD datasets. We retain these behaviors and explicitly annotate both the anomaly type and the recommended recovery action, enabling:
> Robust system development (anomaly detection + recovery)
> Explicit supervision for flow maintenance and error-handling strategies
> (3) A structural mapping mechanism for subjective slots and fuzzy user preferences
> CRSA provides a semantic normalization table and a mapping strategy that converts subjective or vague expressions into structured preference slots paired with the original expression. This addresses the long-standing inability of TOD datasets to model fuzzy or subjective intents, and it forms a foundation for preference reasoning, recommendation strategies, and personalized system replies.
> (4) A transferable human–machine collaborative annotation pipeline
> We introduce a scalable annotation workflow combining human-designed seeds + automatic annotation models + human review, demonstrated in Section 4.1.3 to yield strong accuracy, consistency, and efficiency. The workflow is domain-agnostic and can be transferred to new business areas with minimal seed data and domain knowledge. The methodological contribution lies in this generalizable annotation pipeline, not in the flight-booking topic itself.
> (5) First introduction of controllable personalized system responses in TOD
> CRSA systematically annotates four dimensions of controllable personalization:
> System response style (concise / patient / expert-like)
> Strategy for deviation handling (strict enforcement / flexible adjustment / clarifying jumps)
> Policy for non-business user questions (reject / brief response / patient explanation)
> Dialogue pacing and compactness (rapid progression / user-following rhythm)
> No existing TOD dataset—including MultiWOZ, CrossWOZ, RiSAWOZ—provides such structured controllable personalization signals. Empirically, personalization significantly improves user satisfaction and task success rates. Models trained on CRSA show clear gains in CRAM, task completion, and structural coherence, validating the utility of these annotations.

---

> > ### Author Response · Authors · 2025-11-19
> > **Detailed Rebuttal to Weaknesses 2–3**
> >
> > Response to Weakness 2: Concerning Whether Low Performance Might Be Due to Noise Rather Than Complexity
> > We appreciate this important question and agree that the original manuscript did not fully clarify this point. We offer a more precise explanation below.
> > (1) Section 4.1.1 is a diagnostic stress test—NOT the sole evidence of CRSA’s strength
> > When CRSA is forcibly collapsed into a shallow DST/DA schema, substantial semantic content is lost: multi-turn preference evolution, complex multi-constraint reasoning, subjective slot semantics, anomaly behaviors and recovery logic, system-driven stage transitions.
> > The model is then trained on an oversimplified supervision signal, producing predictably lower scores. These low numbers indicate the limitations of the traditional schema, rather than annotation noise or data deficiencies.
> > (2) The strengths of CRSA are demonstrated collectively across all experiments in Section 4
> > Section 4.1.2: Re-annotating other datasets with CRSA schema improves DST/DA performance → showing the annotation framework generalizes beyond one domain.
> > Section 4.1.4: Ablations show each annotation dimension contributes meaningfully → proving the added complexity is functional, not redundant.
> > Section 4.2: Systems trained on CRSA achieve superior process control, task success, and personalized controllability → demonstrating system-level benefits.
> >
> > Section 4.3: CRSA increases difficulty and discriminability for intent prediction, behavior prediction, slot filling, and user simulation → revealing model capability boundaries.
> > In the revision, we will clearly state that CRSA appears “harder” because it captures realistic semantic richness and structural complexity, not because of noise.
> > (3) Case analysis: traditional TOD schemas cannot represent CRSA-level semantics
> > A real CRSA utterance:“Right then, I’m after the cheapest flight possible tomorrow afternoon—ideally one that’s under the Platinum Card discount cap. I’d even be happy to fly at 1 p.m. if it’s cheap enough. But absolutely no layovers, mind you—unless it’s a quick hop of 40 minutes or less. And since the firm I’m visiting isn’t anywhere near Pudong, I reckon I’ll start by checking flights out of Hongqiao.”
> > This includes conditional logic, evolving preferences, fuzzy time expressions, and multi-objective tradeoffs.
> > Traditional DST collapses this to:
> > time = tomorrow afternoon
> > price = cheap
> > transfer = allowed
> > A severe loss of information. Such complex expressions appear in 37.4% of CRSA turns and 68.3% of dialogues, versus <5% in standard datasets. The difficulty arises from semantic richness, not annotation inaccuracy.
> > (4) Annotation quality is validated by automatic vs. human consistency
> > As detailed in Section 4.1.3, annotations achieve over 90% accuracy, with disagreements mainly at subjective-slot boundaries, not in core structural labels. CRSA’s annotations are stable, coherent, and free from significant noise—the difficulty lies in the input, not in the labels.
> > Response to Weakness 3: Concerning Generalizability and Transferability
> > We fully understand your concerns about generality. Here we clarify: CRSA is not a flight-specific schema, but a general framework for complex transactional TOD.
> > 1. Airline booking is a representative high-complexity transactional scenario
> > Through long-term cooperation with industry partners, we identified that flight booking simultaneously exhibits:multi-slot, multi-constraint, and multi-goal reasoning, highly vague and incrementally revealed user needs, strong contextual dependency and evolving preferences, frequent non-standard behaviors (revisions, backtracking, subtask insertion), real monetary transactions with strict policy constraints.
> > These characteristics appear in hotel booking, e-commerce bundles, insurance consultation, financial planning, and other vertical business domains. Thus, flight booking serves as a representative testbed, not a narrow domain choice.
> > 2. The core contribution is a transferable construction & annotation methodology
> > The methodology addresses:modeling vague and ambiguous expressions, anomaly taxonomy + recovery strategies, subjective slot mapping, stage-based semantic annotation, system behavior triplets, personalization-control labels, human–machine collaborative annotation pipeline.
> > None of these depend on domain-specific features. With new slot definitions and business stages, CRSA’s annotation components transfer directly to other vertical domains.
> > Section 4.1.2 demonstrates: Re-annotating existing TOD datasets using CRSA schema yields significant performance improvements.
> > Section 4.3 further shows: Models trained on CRSA exhibit stronger diversity, structure, and contextual adaptability—abilities that are inherently task-agnostic and transferable.
> > Thus, CRSA complements existing multi-domain datasets by providing a high-fidelity blueprint for modeling complex business semantics.

---

> > > ### Author Response · Authors · 2025-11-19
> > > **Response to Questions Q1–Q3**
> > >
> > > Q1. Contribution of Different Data Sources to the Dataset and Task Performance
> > > We sincerely appreciate this important question. The current version of the paper focuses primarily on the overall effectiveness of the dataset and the construction pipeline, and we agree that the contribution of each data source deserves a more explicit and fine-grained discussion. In the revised manuscript, we will add detailed analyses. Conceptually, the three data sources play distinct roles:
> > > Real business dialogues provide authentic user behavior patterns and naturally occurring anomaly types, serving as the empirical foundation for defining anomaly categories, recovery strategies, and subjective-slot mapping guidelines. They function as the semantic and procedural anchor of the dataset.
> > > Crowdsourced role-playing dialogues, conducted under real business rules and guided by structured protocols (Appendix A.1 & A.2), enable systematic expansion of dialogue trajectories, anomaly combinations, and preference-evolution patterns. They supplement realistic but underrepresented scenarios that seldom appear in raw business logs.
> > > LLM-generated dialogues, produced using few-shot prompts and domain constraints, enrich long-tail combinations, complex multi-constraint compositions, and iterative preference-revision cases. This enhances semantic variety, reinforces subjective-slot expression, and increases structural diversity.
> > > In our existing experiments, we primarily reported the integrated effect of “multi-source aggregation + standardization + structured annotation.” In the revision, we will also provide more granular analysis, including:
> > > (1) Dataset Composition
> > > The final CRSA dataset contains 1,480 dialogues (26,482 turns), with the following distribution:
> > > Real business dialogues: 372 (≈ 25.1%)
> > > Crowdsourced role-playing dialogues: 808 (≈ 54.6%)
> > > LLM-augmented dialogues (GPT-4o): 300 (≈ 20.3%)
> > > (2) Complexity Statistics of Each Source
> > > Real dialogues: avg. turns = 18.4; anomaly rate = 39.2%; subjective-slot mapping = 43.5%; stage backtracking = 26.2%; subtask insertion = 28.2%.
> > > Crowdsourced dialogues: avg. turns = 24.1; anomaly rate = 31.6%; subjective-slot mapping = 40.6%; stage backtracking = 27.9%; subtask insertion = 33.3%.
> > > LLM-generated: avg. turns = 17.2; anomaly rate = 25.3%; subjective-slot mapping = 37.6%; stage backtracking = 19.5%; subtask insertion = 26.8%.
> > > (3) Source-wise Training Comparison
> > > To quantify each source’s specific contribution to system performance, we trained models using each subset independently (under identical architectures and configurations) and compared them with the full three-source dataset (CRSA). Results:
> > > | Training Source          | DA Acc. | ADFC | CRAM | TCR  |
> > > | ---------------------------- | ----------- | -------- | --------- | ------- |
> > > | Real Only                   |   86.3     |   82.6  |  87.3   | 86.2  |
> > > | Crowdsourced Only   |   90.6     |   92.8  |  85.6   | 81.5  |
> > > | LLM Only                   |   88.4     |   86.7  |  91.2   | 88.7  |
> > > | Full CRSA                  |   92.4     |   94.0  |  93.8   | 90.9  |
> > > The three-source integration clearly yields the best performance across task completion, process control, personalization alignment, and generation diversity. These results demonstrate the complementary roles of the three data sources and their combined value in constructing a robust TOD corpus.
> > >
> > > Q2. Code Repository and Availability
> > > We have updated and released the full repository, including dataset samples, annotation specifications, and automatic annotation models:https://anonymous.4open.science/r/CRSA-CBBB
> > > This ensures full reproducibility and transparency for evaluation and further research.
> > >
> > > Q3. Human Cost of Preparing Seed Data for Automatic Annotation
> > > To train the automatic annotation model, we collected approximately 500 real business dialogues and performed manual cleaning, filtering, and annotation.
> > > In total, 409 dialogues were annotated, comprising 7,526 turns, requiring roughly 32 hours of human labor.
> > > This relatively modest annotation cost produced an automatic annotation model achieving over 86% field-level agreement, substantially lower than the cost of building a large TOD dataset from scratch. These empirical costs demonstrate the transferability and practicality of our human–machine collaborative annotation methodology.

---

### Official Review · Reviewer_zWUc · 2025-11-10

**Soundness:** 2
**Presentation:** 2
**Contribution:** 3
**Rating:** 6
**Confidence:** 2

**Summary:**

The paper proposes a new dataset for task oriented dialogue with multiple semantic annotations.  The quality of the dataset is demonstrated by comparing with existing TOD datasets and the potential applications of the dataset are shown on four canonical TOD subtasks.

**Strengths:**

A new dataset with rich semantics is proposed for Chinese airline booking scenario.

Extensive experiments are conducted to verify the effectiveness or advantages of the dataset compared with the previous datasets.

Ablations of the rich semantic annotations and the automated compared with manual annotation are provided to show the effectiveness of the data annotation pipeline.

**Weaknesses:**

**Not very self-contained presentations.** The dataset proposed in the paper is about the air line booking scenario and one of the major contribution is about rich semantic annotation. Although the paper provides many statistics about the dataset compared with existing ones, no examples to show the typical booking dialogue and the claimed ''rich semantic annotation''. In the main paper, no example dialogue is demonstrated and only very limited examples are mentioned in the supplemental material.

**Lack of clarification on the comparisons with existing TOD dataset.** The datasets are collected for different tasks, for example, the proposed dataset for airline booking while crossWOZ for transportation, tourism, etc. **How to compare these dataset in a fair way? Does the questions for the comparison set for the airline booking? This seems unfair comparisons with existing datasets?**

**Questions:**

see the weakness above

---

> ### Author Response · Authors · 2025-11-19
> **Addressing Self-Containment and Comparative Evaluation Concerns**
>
> We sincerely appreciate your careful reading of our work and your positive comments. Below we respond to your two main concerns and explain how we plan to enhance the revised manuscript.
> 1. On the concern the paper is not sufficiently self-contained / lacks typical dialogue examples
> We fully agree with your observation. For a dataset whose key contribution lies in rich semantic annotations and structured modeling of dialogue processes, statistical tables and abstract descriptions alone are insufficient for readers to intuitively understand the characteristics of the data and its annotation design. In the current version, full examples appear mainly in the appendix, which indeed reduces the self-contained nature of the main body. This is a shortcoming in our writing.
> To address this, in the revised manuscript we will:
> Add a complete representative dialogue + multi-layer annotations example directly in the main paper.
> The example will be carefully selected to contain simultaneously:Vague expressions and gradual clarification of preferences;Non-standard user behaviors (e.g., vague, altering, irrelevant, or contradictory turns);System-driven process control and strategy adjustment.
> And we will present all corresponding structured annotations, including:Dialogue stage, system behavior type, user semantic analysis (including anomaly type when present), personalized control tags,structured links between recommended candidate options and user choices, slot filling, subjective preference mapping, and slot normalization results.
> This addition will allow readers to understand the dataset’s multi-dimensional semantic structure without needing to read the appendix, significantly improving self-containment and interpretability.
> 2. On the concern whether cross-dataset comparison is fair across tasks/domains
> 2.1 The comparisons do not aim to claim CRSA is superior to all existing TOD datasets.
> Rather, our goal is to demonstrate that CRSA provides:
> A different kind of challenge, particularly around flow control, non-standard behaviors, and preference evolution,
> A complementary evaluation perspective to multi-domain TOD datasets,under identical models and training setups.
> Existing datasets (e.g., MultiWOZ, CrossWOZ) are designed primarily to showcase wide domain coverage and cross-domain generalization.In contrast, CRSA focuses on a single but highly complex transactional scenario—characterized by:multi-slot, multi-constraint, multi-objective reasoning,strong contextual dependencies,real monetary and policy constraints,user goals that evolve and require multiple clarifications,high frequency of non-standard user behaviors.
> Thus, the purpose of our experiments is not to claim that CRSA “outperforms” these datasets, but rather to show that under the same model settings, CRSA exposes distinct modeling challenges and allows evaluation of abilities that existing datasets do not explicitly support.
> In the revision, we will clearly articulate this positioning: CRSA complements, rather than competes with, existing multi-domain datasets by offering a testbed for controllable, process-guided TOD in complex transactional settings.
> 2.2 Importantly: all datasets are evaluated under their own task definitions—not converted to airline booking.
> To clarify:We did not convert any dataset into airline booking, nor did we impose CRSA’s task setting onto other corpora.Each dataset is trained and evaluated strictly under its original domain and task schema,Only the input–output format and training configuration of the model are kept consistent.The comparison therefore assesses:How the same model performs across datasets that differ in their intrinsic difficulty, structure, and semantic complexity.
> We will explicitly state this methodology in the revised paper to prevent any misunderstanding.
> 2.3 Additional clarification on fairness and refined wording
> We agree that cross-domain comparisons can inherently raise fairness concerns.Thus, we will soften or revise any wording in the paper that could be interpreted as claiming that CRSA is “more complex” or “superior.”
> In the revision, we will instead state: CRSA’s linguistic and process complexity—stemming from ambiguity, preference drift, non-standard behaviors, and system-driven flow control—offers a different and complementary difficulty profile to existing multi-domain datasets.Multi-domain datasets are excellent for evaluating breadth and cross-domain generalization,
> Whereas CRSA is designed for evaluating process control, exception robustness, and controllable system behavior in complex transactional dialogues.
> We will emphasize: CRSA is not a replacement for existing flight-related corpora nor for multi-domain datasets; rather, it systematically models semantic phenomena that are currently missing and needed in real-world transactional TOD.

---

### Official Review · Reviewer_iv5x · 2025-11-10

**Soundness:** 1
**Presentation:** 2
**Contribution:** 1
**Rating:** 2
**Confidence:** 4

**Summary:**

This paper proposed a Chinese single-domain task-oriented dialogue dataset, focusing on flight booking. The dataset is composed of three different sources, including real dialogues, crowd-sourced dialogues, and LLM-assisted generated dialogues.

**Strengths:**

* A single-domain TOD dataset is proposed.

**Weaknesses:**

* The definition of the question this work aimed to tackle should be clarified. At the end of Section 2, Related Work, the authors mentioned that previous works cannot capture dynamic scenarios, model anomalous user expression and support system-driven dialogue guidance. However, it is unclear how the proposed dataset captures dynamic scenarios, especially since it is a single-domain dataset without domain switching or coreference. In addition, why are the unclear, vague, or irrelevant user behaviour defined as "anomaly" behaviour and filtered out during data cleaning (see Section 3.2), when they are common in real-world interactions? Furthermore, what is the definition of "system-driven" dialogue? Does it refer to the proactive dialogue system or something else?
* Following the previous point, what is the definition of "limited semantic diversity" and "insufficient process control"? Without a clear definition of these points, it is difficult to estimate the contribution of this work, especially since there are plenty of flight booking datasets which is more complex than this proposed dataset [1,2].
* In addition, the novelty of multisource data integration, structural standardisation, and hierarchical annotation framework should be elaborated further. Various TOD datasets also include dialogues from different sources, e.g. AirDialogue [2] and EmoWOZ [3], in order to enrich the variety of the dataset and to bridge the gap between human-to-human and human-to-machine conversations. Unified format across corpora is also proposed previously [4], and the hierarchical annotation framework, context-dialogue-slot, is a standard annotation schema, which is also presented in MultiWOZ, SGD, crossWoZ, etc.
* The effectiveness of the experiments should be clarified as well. For example, the authors claimed that their proposed dataset exhibits greater semantic richness and diversity, as evidenced by the lower performance of mBART for DST compared to the other datasets in Table 3. However, it is unclear why lower performance on DST means the dataset is more diverse, which should be supported by other studies or more experiments.


[1] Frames: a corpus for adding memory to goal-oriented dialogue systems (El Asri et al., SIGDIAL 2017)

[2] AirDialogue: An Environment for Goal-Oriented Dialogue Research (Wei et al., EMNLP 2018)

[3] EmoWOZ: A Large-Scale Corpus and Labelling Scheme for Emotion Recognition in Task-Oriented Dialogue Systems (Feng et al., LREC 2022)

[4] ConvLab-3: A Flexible Dialogue System Toolkit Based on a Unified Data Format (Zhu et al., EMNLP 2023)

**Questions:**

* The citation of MultiWOZ is incorrect (L081). It should be [1], instead of [2], which is MultiWOZ 2.4.
* In the related work, the authors mentioned that pipeline systems are facing joint optimisation issues, which is not always true [3].
* What is the meaning of "complex business scenarios" (L054)?
* Statistically significant test is missing, making it difficult to assess the results, especially Table 3, 4, 6, 7, and 9.


[1] MultiWOZ - A Large-Scale Multi-Domain Wizard-of-Oz Dataset for Task-Oriented Dialogue Modelling (Budzianowski et al., EMNLP 2018)

[2] MultiWOZ 2.4: A Multi-Domain Task-Oriented Dialogue Dataset with Essential Annotation Corrections to Improve State Tracking Evaluation (Ye et al., SIGDIAL 2022)

[3] A Generative Model for Joint Natural Language Understanding and Generation (Tseng et al., ACL 2020)

---

> ### Author Response · Authors · 2025-11-19
> **Response to Conceptual Clarifications**
>
> Dear Reviewer, We sincerely thank you for the thorough reading of our work and for the concrete suggestions you provided. Below we offer a detailed and structured response. Corresponding revisions will be incorporated in the final manuscript.
> I. Clarifying Research Objective & Key Concepts
> 1.1 Research Problem
> This work targets three phenomena that frequently appear in real Chinese transactional TOD but are insufficiently supported by existing datasets: fuzzy, incomplete, contradictory, or subjective user expressions; semantics that depend heavily on historical context, with user goals and preferences evolving dynamically across turns; dialogue disruptions caused by unconventional user behaviors, such as off-topic turns, reversals, repeated revisions, clarifications, or subtask insertions.
> Airline booking serves as a representative scenario for validating a generalizable annotation and construction methodology, not as a domain-specific contribution.
> 1.2 Meaning of Dynamic Scenarios
> Our intended meaning refers specifically to dynamics internal to a single business domain, including: continual supplementation, revision, or reversal of user goals and constraints; progressive revelation and fluctuation of preferences driven by candidate comparison; backtracking, jumps, subtask insertion, interruptions, and subsequent recovery at the dialogue-flow level; system-initiated adjustments to stages, re-entry into sub-flows, re-clarification of critical slots, and continual maintenance of stage progression and transaction consistency.
> In the revision, we will replace the ambiguous term “dynamic scenarios” with the more precise formulation: Dynamic in-domain conversational scenarios in which user goals, constraints, and preferences evolve and are revised across multiple turns. We will further support this with statistics and illustrative examples.
> 1.3 Misunderstandings Regarding Anomaly
> We fully understand your concern that the manuscript may have implied that phenomena such as unclear / vague / irrelevant utterances were removed during data cleaning. This is not the case. The key clarification is:
> In Section 3.2, we remove only entire dialogues that are globally unusable, such as those with severe transcription errors, entirely off-task content, or missing critical turns rendering the business flow unrecoverable.
> In Section 3.3, we intentionally preserve and explicitly annotate all local anomalies that occur within otherwise usable dialogues. These include the six anomaly types: Unclear, Vague, Irrelevant, Alter, Error, Default. We annotate their category, cause, and recommended system recovery strategy.
> Thus, real-world behaviors such as ambiguity, vagueness, off-topic turns, or reversals are not removed as noise, but are exactly the kinds of phenomena we aim to model. They are systematically categorized and annotated as part of our anomaly analysis, forming an essential component of CRSA’s design.
> In the revision, we will revise the phrase semantically unclear in Section 3.2 to whole-dialogue unintelligible or entirely off-task.
> We will also add an explicit statement at the start of Section 3.3: It is important to clarify that user-side phenomena such as vagueness, ambiguity, off-topic turns, and revisions are not removed during data cleaning. Instead, they are explicitly annotated and modeled through the anomaly analysis framework introduced in this section.
> 1.4 Definition of System-Driven Dialogue
> To avoid ambiguity, we will formally define system-driven dialogue in the revised manuscript. In our work, it refers not merely to generic “proactive systems,” but to a structured and operationally defined paradigm:
> Flow level: The dialogue follows three explicit stages (Information Collection → Candidate Recommendation & Comparison → Supplemental Clarification & Transaction Confirmation). Stage transitions are system-initiated, based on tracked user state and task progress—rather than solely reacting to the user. System behaviors proactively address bottlenecks—such as unclear descriptions, need for recommendations, or user misunderstandings—through targeted questioning, suggestions, or stylistic adjustments.
> Behavior level: We define 63 system behaviors, each mapped to a specific stage, forming a structured stage × behavior-type strategy space. We also provide four-dimensional personalization-control labels (style, deviation-handling, task-irrelevant query policy, and flow tightness), enabling distinct stylistic executions of the same strategy.
> Thus, in the revision, we will define system-driven dialogue as: A dialogue paradigm in which the system takes primary responsibility for driving business process progression—via explicit stage transitions, behavior selection, and four-dimensional control signals—and these mechanisms are modeled through structured stage annotation, behavior triplets, and personalization tags.

---

> > ### Author Response · Authors · 2025-11-19
> > **Response on Semantic Diversity, Process Control, and Relation to Existing Flight-Booking Datasets**
> >
> > 1. Definition of Semantic Diversity
> > We agree that in the current manuscript, the notion of “limited semantic diversity” is insufficiently concrete. In the revised version, we will refine this concept into several measurable dimensions, including:
> > User expression level: proportion of vague/subjective expressions, higher frequency and variety of anomaly types;
> > Slot and intent level: larger slot-value space, higher proportion of subjective slots, greater frequency of multi-turn revisions and conflicts;
> > System behavior and response level: richer distribution of system behavior types (rather than fixed template cycles), higher Distinct-n and lower Self-BLEU, indicating reduced template bias and greater linguistic variability.
> > We will emphasize that a lower DST score alone cannot be interpreted as “greater diversity.” In the revision, DST/DA/BLEU metrics will be used only in combination with statistical indicators and system-level experiments as collective evidence that CRSA poses higher semantic and behavioral modeling challenges for TOD systems.
> > 2. Definition of Process Control
> > In our work, process control refers to:“A model’s ability to select the appropriate next stage and the correct system action based on contextual state, ensuring stable task completion within limited turns.”
> > We quantify this capability using structural metrics such as ADFC, TCR, STA, and end-of-dialogue timing accuracy. In the revised manuscript, we will provide a more concise and formal definition, and replace the vague phrase “insufficient process control” with:“Existing TOD datasets generally lack explicit stage annotations, system behavior labels, and control signals, leaving models with no structural supervision for process-control metrics (e.g., ADFC, STA, FCC). CRSA is deliberately designed to provide this supervision.”
> > 3. Relation to Existing Flight-Booking Datasets
> > You noted that datasets such as Frames and AirDialogue contain complex airline-booking interactions. In the revised manuscript, we will make explicit that CRSA is not intended to compete with these datasets in terms of “complexity” or scenario breadth. Rather, CRSA targets a different class of real-world challenges, specific to Chinese transactional dialogue:
> > vague, ambiguous, or subjective user expressions;unstable or evolving goals;spontaneous subtask insertion;anomaly behaviors requiring structured recovery;system-driven flow control and strategy selection.
> > The semantic challenges CRSA addresses are not overlapping with those in Frames or AirDialogue. CRSA contributes a transferable construction framework for complex business dialogues, consisting of: stage-transition modeling, system-strategy control space, fine-grained user semantic analysis, multi-dimensional personalization labels, unified task-state representation.
> > Compared with existing datasets, CRSA supports anomaly modeling, controllable personalization, system-driven strategy planning, interpretable process reasoning, multi-task auto-annotation, and flow-recovery evaluation—capabilities not covered elsewhere.
> > Thus, the revised manuscript will emphasize: The value of CRSA lies in its methodology, not in the airline domain itself.
> > CRSA provides a structured annotation framework for high-frequency yet under-modeled phenomena in real Chinese transactional scenarios, serving as a complement—not a replacement—to existing TOD corpora.

---

> > > ### Author Response · Authors · 2025-11-19
> > > **Response on Innovation and Interpretation of Experimental Results**
> > >
> > > Addressing Concerns Regarding Limited Novelty
> > > We agree that multi-source data construction, unified formatting, and a context–dialogue–slot hierarchical structure are not new concepts. We will explicitly distinguish prior work from CRSA’s new contributions.
> > > The contribution of CRSA is not in creating yet another airline-booking dataset, but in proposing a transferable deep semantic annotation methodology for complex business-oriented TOD scenarios. For the first time, CRSA adopts a stage-based process-semantic framework as its central design principle, decomposing business workflows into learnable structures involving: stage-level semantic representations, an expanded inventory of system actions, candidate–user decision dependencies, causal linkages between information gathering, option comparison, and decision-making.
> > > This structured design is domain-agnostic—only slot definitions and business rules need to be substituted to port the annotation schema to other complex verticals—forming a reusable component library for process-controlled TOD annotation.
> > > CRSA also systematically models real-world user anomaly behaviors and annotates recovery strategies conditioned on stage logic and system-behavior labels. Unlike existing datasets, which often filter or downplay digressions, inconsistencies, reversals, conflicting constraints, or underspecified goals, CRSA explicitly preserves them. This enables models to learn how to maintain process correctness under noisy, unstable input—an essential capability for real-world deployment.
> > > Additionally, CRSA introduces a structured mapping mechanism for subjective and fuzzy preference expressions, converting natural utterances such as “as cheap as possible,” “not too early,” or “anything should be fine” into learnable preference constraints while retaining their natural-language forms. This dual representation lets models correctly interpret vague semantics and utilize structured constraints during decision-making.
> > > CRSA is the first TOD dataset to systematically incorporate controllable personalization, encoding system style, flow pacing, deviation-handling strategy, and policy for task-irrelevant questions as explicit control signals. These behavioral dimensions are crucial in real commercial systems yet almost entirely absent in existing corpora. Experiments show that these control labels significantly improve process coherence, task success rate, linguistic diversity, and contextual alignment. They also provide natural supervision signals for reinforcement learning and policy optimization, pushing TOD research beyond mere task completion toward controllable, reliable, and user-aligned task completion.
> > >
> > > Clarifying the Interpretation of Experimental Results and “Low DST Performance”
> > > You correctly noted that using mBART’s low DST performance as direct evidence of semantic richness is not theoretically justified. We fully agree. In the revised manuscript, we will tighten the argument with the following refined interpretation:
> > > 4.1.1: When CRSA is forcibly compressed into a traditional DST/DA schema, dynamic preferences, fuzzy expressions, anomaly behaviors, context dependencies, and stage-driven control signals are inevitably collapsed or erased. The model performs poorly under this oversimplified schema, indicating that traditional shallow annotation formats are insufficient to capture CRSA’s semantic phenomena, rather than reflecting dataset noise or lower quality.
> > > 4.1.2: Re-annotating existing datasets (e.g., RiSAWOZ, CrossWOZ) using the CRSA schema significantly improves DST/DA/NLG metrics, demonstrating that CRSA’s annotation dimensions provide additional useful structure—even when the underlying dialogues remain unchanged.
> > > 4.1.4: Ablation studies show that each CRSA annotation layer (stages, anomalies, candidate structures, personalization labels) makes a substantive contribution. These are not “complex but useless” additions.
> > > 4.2–4.3: Models trained on full CRSA annotations exhibit strong capabilities in process control, anomaly handling, personalized generation, and interpretable strategy decisions—abilities that traditional TOD datasets cannot measure or supervise. These experiments show that CRSA helps explore model capability boundaries and provides meaningful signals for analytical evaluation.
> > > Thus, in the revised manuscript we will replace the earlier phrasing with the more accurate formulation:
> > > “Under the same model and training setup, CRSA yields lower performance in unified DST/DA schemas. Combined with statistical evidence on anomaly frequency, goal/constraint modifications, linguistic variability, and flow complexity, this indicates that CRSA presents substantially higher modeling challenges. Its value lies in enabling fine-grained supervision and evaluation for complex real-world scenarios.”

---

> > > > ### Author Response · Authors · 2025-11-19
> > > > **Responses to Reviewer Questions and Planned Revisions**
> > > >
> > > > Q1. Correction of MultiWOZ Citation
> > > > We will correct the citation as suggested and use Budzianowski et al., 2018 as the canonical reference for MultiWOZ.
> > > > Q2. Clarification on Pipeline Joint Optimization
> > > > We will revise the statement and cite works such as Tseng et al., 2020, specifying that while traditional modular pipelines often face challenges in joint optimization, generative modeling approaches have been explored as potential mitigation. This revision avoids overgeneralization.
> > > > Q3. Definition of “Complex Business Scenarios”
> > > > In the revised manuscript, we will explicitly define complex business scenarios as tasks exhibiting the following characteristics:
> > > > 1. Multi-slot, multi-constraint, multi-goal interactions: Strong coupling among heterogeneous slots such as time, price, routing, fare rules, and passenger attributes.
> > > > 2. Highly vague and incomplete user needs: Requirements are progressively revealed across multiple turns rather than provided upfront.
> > > > 3.Strong contextual dependency and preference evolution: Budget, timing, and routing preferences often shift dynamically as candidate options are presented.
> > > > 4. Frequent non-standard user behaviors: Including option backtracking, need revisions, subtask insertion, and off-task queries.
> > > > 5. Real monetary transactions and compliance constraints: Involving fare computation, price differences, rule-based restrictions, and user-risk considerations.
> > > > We will also clarify why airline booking serves as a representative and challenging testbed: it simultaneously involves high financial stakes, strict constraints, multiple candidates, multi-turn comparisons, and vague and evolving user preferences, making it structurally more complex than typical TOD domains such as e-commerce or travel planning.
> > > >
> > > > Q4. Statistical Significance Testing
> > > > We will incorporate concise but sufficient significance testing for all core evaluations. Specifically:
> > > > 1. Statistical methods (consistent with established practice in TOD research):
> > > > · Paired bootstrap resampling (1,000 iterations): for sequence-based metrics (BLEU, ROUGE, CRAM).
> > > > · Paired t-tests (α = 0.05): for scalar metrics (DST, DA, Accuracy, TCR, ADFC, etc.).
> > > > 2. Tables to be updated with significance markers:
> > > > · Table 3: DST / DA
> > > > · Table 4: Improvements from re-annotation
> > > > · Table 7: Process-control metrics (ADFC, TCR, STA)
> > > > · Table 9: Semantic-diversity metrics
> > > > These revisions will ensure statistical rigor and strengthen the empirical findings.

---

> > > > > ### Comment · Reviewer_iv5x · 2025-11-26
> > > > >
> > > > > Thank you for your response. As it requires significant revision and the new revision has not been submitted yet, I would like to keep my score unchanged.

---

### Official Review · Reviewer_JxY4 · 2025-11-11

**Soundness:** 3
**Presentation:** 3
**Contribution:** 3
**Rating:** 6
**Confidence:** 3

**Summary:**

The paper introduces CRSA, a Chinese TOD dataset in the airline booking domain that contains diverse sources (real, crowd-sourced, and LLM-generated) and rich annotations.

The paper conducts extensive experiments to evaluate the quality and usefulness of CRSA. Results show that transforming other datasets into CRSA schema significantly improves the effectiveness of those datasets. Ablation studies also show that removing any of the components will harm the metric scores, validating the effectiveness of each component.

**Strengths:**

1. CRSA contains >1400 dialogs and >26000 turns, with more system-led control and diverse user behaviors, making it larger than existing baselines, and more diverse and representative of real-world scenarios.

2. CRSA also contains rich and process-aware annotation. It incorporates three-tier schema, user anomaly modeling, system behavior triplets, fuzzy expression handling, etc. Ablation studies show that removing any component hurts the scores, validating the effectiveness of each piece.

3. Re-annotation gains on other datasets demonstrate the strong utility of the annotation schema.

**Weaknesses:**

1. Focusing on only one domain (airline booking) constrains cross-domain generalization and limits application to broader scenarios.

2. CRSA scores the lowest on DST/DA/BLUE/ROUGE-L after being converted to a common schema. While interpreted as "harder", it also signals that current models struggle to learn and generalize on CRSA effectively. It doesn't seem a fair comparison and does not reflect the advantage of CRSA.

**Questions:**

1. What are the portions of real, crowdsourced, and LLM-generated dialogues in the final corpus?

2. For LLM-generated dialogues, could you share more details on data quality judgment and filtering?

---

> ### Author Response · Authors · 2025-11-19
> **Clarifications on Novelty, Interpretation of Model Performance, and Methodological Generalization**
>
> We sincerely thank Reviewer for the careful reading of our work and for recognizing the dataset scale, system-driven dialogue structure, multi-source construction, rich process-aware annotation scheme, and ablation effects. Below we provide a more detailed and focused response to each Weakness and Question.
>
> Response to Weakness 1
> Here, we clarify that our choice of airline ticket booking as a single domain is a deliberate design for deep, high-difficulty validation, rather than a methodological restriction.
> Airline booking is a representative case of complex business-oriented TOD. Based on long-term collaboration with industry practitioners, we observed that it simultaneously involves (i) tightly coupled multi-slot, multi-constraint, multi-goal interactions (ii) user needs that are inherently vague, incomplete, and gradually revealed across multiple turns, (iii) strong contextual dependency and preference evolution driven by candidate comparison, (iv) frequent non-cooperative behaviors (goal switching, option backtracking, insertion of sub-queries), and (v) real monetary transactions and compliance constraints. These characteristics are highly aligned with other vertical domains such as hotel booking, insurance consultancy, financial product selection, and bundled e-commerce services. Thus, airline booking serves as a prototypical testbed, rather than a narrow domain.
> More importantly, the core contribution of CRSA lies in a transferable data construction and annotation methodology, not in the domain itself. Our work addresses three key challenges:
> • modeling vague expressions, semantic ambiguity, and unconventional user behavior;
> • explicitly encoding dialogue structure, business flow, and behavioral strategies through staged process annotation, anomaly analysis, subjective-slot mapping, behavior triplets, and personalized control labels;
> • scaling deep annotation via a human-in-the-loop+multi-task auto-annotation pipeline.
> None of these designs are domain-dependent. Once slot definitions and process stages are redefined for a new vertical, our slot-mapping strategy, process-control modeling, and stage-aware semantic decomposition can be directly reused. Section 4.1.2 already shows that re-annotating existing TOD datasets using the CRSA schema significantly improves DST/DA performance, demonstrating that our annotation framework benefits other corpora as well.
> CRSA deliberately deepens structural modeling of vague expression, anomaly behaviors, process control, and personalized strategies in a single complex domain. It provides a high-fidelity blueprint for replicating this methodology in other verticals. We will make this positioning clearer in the revised version.
>
> Response to Weakness 2
> We agree that the current manuscript does not adequately clarify the interpretation of the “low performance” phenomenon. We provide a more consolidated explanation below.
> The experiment in Section 4.1.1 is designed as a diagnostic stress test, aimed at exposing the limitation of traditional shallow DST/DA schemas rather than serving as the sole evidence of CRSA’s advantages. When CRSA is forcibly compressed into the conventional DST/DA label space, most key semantics (multi-turn preference evolution, complex constraint composition, subjective-slot interpretation, anomaly behaviors and recovery strategies, system-driven stage progression) are inevitably lost. As a result, the model learns only from a reduced semantic view and naturally shows lower performance. The lower scores therefore reflect that the traditional schema lacks the expressiveness needed to cover CRSA’s semantic complexity, not that CRSA data is noisy or of low quality.
> CRSA’s benefits are demonstrated through the entire Section 4, not through one experiment:
> • Section 4.1.2 shows that re-annotating other datasets with the CRSA schema consistently improves DST/DA metrics, proving the annotation framework is beneficial beyond CRSA.
> • Section 4.1.4 shows that removing any annotation dimension (stage, anomaly, options, subjective slots) significantly degrades performance, demonstrating that each component carries meaningful semantic supervision.
> • Section 4.2 reveals that models trained on CRSA gain clear advantages in process control, task completion, and controllable personalized generation—capabilities that conventional datasets cannot exercise.
> • Section 4.3 further shows that CRSA increases task difficulty and discriminatory power across intent classification, action prediction, slot filling, and user simulation.
> In the revised version, we will explicitly avoid the phrasing “low score = better,” and instead clarify: Under a unified shallow-label schema, performance degradation aligns with statistical observations of CRSA’s high proportion of ambiguity, preference evolution, complex constraints, and non-linear flows. This indicates that richer semantics require finer-grained annotation to be fully modeled.

---

> > ### Author Response · Authors · 2025-11-19
> > **Multi-Source Contributions, Data Quality, and Annotation Costs**
> >
> > Response to Question 1
> > The final CRSA corpus contains 1,480 complete dialogues, composed of:
> > • Real business logs: 372 dialogues (≈25.1%)
> > • Crowdsourced dual-role simulations: 808 dialogues (≈54.6%)
> > • LLM-augmented dialogues (GPT-4o): 300 dialogues (≈20.3%)
> > We intentionally controlled distributional differences across sources (avg. turns, anomaly rate, subjective-slot frequency, stage backtracking patterns), ensuring that multi-source integration enhances diversity without distorting real behavioral patterns. Real logs provide the baseline distribution of authentic behaviors; crowdsourced dialogues systematically fill coverage gaps in valid business flows; LLM data enriches long-tail combinations and rare constraint interactions.
> >
> > Response to Question 2
> > We understand your concern regarding the quality of synthetic data. The LLM-generated portion (20.3%) is used to expand long-tail complex scenarios on top of the real + crowdsourced backbone. Its quality is ensured via a three-layer control pipeline:
> > (1) Structured prompting constraints: LLMs must produce dialogues that follow realistic three-stage flow, include progressive information revelation, contain 1–2 anomaly behaviors with reasonable recovery, and maintain natural tone and domain-specific style.
> > (2) Automated structural and logical filtering: We use an LLM-based validator to check turn alternation, stage consistency, slot coherence, pricing plausibility, anomaly–recovery alignment, and system-driven flow control. Samples failing any check are discarded; borderline samples enter manual review.
> > (3) Human auditing and style alignment: Human reviewers examine naturalness, domain compliance, and stylistic consistency, ensuring that the retained LLM samples align with real/crowdsourced distributions rather than drifting into artificial patterns.
> > The remaining LLM samples exhibit process structures and linguistic naturalness comparable to human-generated dialogues. Additional examples will be added to the appendix in the revised version.

---

### Author Response · Authors · 2025-12-02
**Revision Summary**

We sincerely thank the reviewers and AC for the time, thoughtful analysis, and constructive feedback provided throughout the review process. We carefully examined all raised concerns and have now substantially revised the manuscript accordingly. The updated version reflects both conceptual clarification and methodological strengthening, as well as more rigorous experimental validation.

Specifically, we have made the following major revisions:
1.Clarified Positioning in Related Work.
We refined the discussion on pipeline-based TOD systems to avoid overly absolute claims. We additionally incorporated work such as Tseng et al., 2020, which demonstrates that unified latent policy modeling and generative training strategies may alleviate joint optimization challenges. We also extended the section by clearly articulating the remaining gaps that CRSA addresses.
2.Formal Definition of Complex Business Scenarios.
A concise overview was added in the Introduction, and Section 3.1 now provides a formal and well-grounded definition supported by prior TOD literature. We also further justified the selection of the airline booking domain as a canonical representative of such scenario types.
3.Statistical Significance Validation.
We added paired bootstrap resampling and paired t-tests across core metrics. Corresponding results and indicators now appear in Tables 3, 4, 7, and 9.
4.Clarified Novelty Contribution.
Section 3 was expanded to explicitly separate existing conventions from CRSA's newly introduced modeling dimensions, including fuzzy preference mapping, anomaly simulation and recovery supervision, personalization control, and the transferable methodology behind the annotation framework.
5.Improved Experimental Explanation.
We revised Section 4.1.1–4.1.4 to more accurately interpret results. The focus is now:“Traditional shallow schemas are insufficient to express the semantic phenomena present in CRSA, and the proposed annotation framework provides additional useful structure, enabling deeper reasoning and process control.”
We adjusted wording to avoid misleading causal inference (e.g., “low DST → higher complexity”).
6.Clarified Data Cleaning Policy.
We refined Section 3.2 to state that only globally unusable samples (e.g., irreparable transcription errors, missing stages) were removed, while all realistic divergence behaviors were preserved and annotated.
7.Expanded Dataset Characteristics and Statistics.
Section 3.4 now contains detailed breakdowns of data composition, scenario coverage, linguistic diversity, and semantic characteristics.
8.Added Missing Citations and Differentiation.
We included the references mentioned by reviewers and clarified conceptual distinctions between CRSA and prior datasets (e.g., Frames, AirDialogue).
9.Strengthened Data Generation Workflow Description.
Appendix B.3 now details automated and human verification workflows, quality filters, and bias control procedures.
10.Added Cross-Domain Evaluation.
A new section in Appendix F reports transferability experiments demonstrating how CRSA’s annotation methodology generalizes beyond the airline domain.
11.Expanded Annotation Example and Schema Demonstration.
Appendix E now includes a complete, stage-aware annotated example aligned with the full CRSA schema.

We deeply appreciate the reviewers’ and AC’s insightful feedback, which directly contributed to improving the clarity, rigor, and framing of the work. We respectfully invite you to refer to both the updated manuscript and earlier rebuttal responses for full details.
Thank you again for your careful evaluation and valuable guidance.

---

### Meta-Review · Area_Chair_i61v · 2026-01-07

**Summary:**

The paper introduces a Chinese task-oriented dialogue dataset, CSRA, designed to address limitations of existing TOD data. CRSA integrates diverse data sources to create realistic dialogues and employs a multi-level annotation framework capturing dialogue acts, user intents, and task flows. Extensive experiments show that CRSA improves data quality, system training effectiveness, and task adaptability, making it a strong resource for process modeling, strategy learning, and response generation in TOD systems. The dataset will be made publicly available and an anonymized repo is provided as supplementary data with the paper.

**Reviewer Concerns:**

Reviewers are concerned about limited novelty of the work in the paper as well as single domain being a limitation. Rebuttals tried to answer these concerns, but I agree with one of the reviewers that the responses are not sufficient. It almost feels as if the authors are confusing multi-domain dataset and multi-domain dialogue.

**Reviewer Scores:**

I think the reviewers wouldn't change their scores in response to rebuttals (one reviewer even says so).

---

### Decision · Program_Chairs · 2026-01-26

Reject